# A class of extracellular vesicles from breast cancer cells activates VEGF receptors and tumour angiogenesis

Qiyu Feng[1,†], Chengliang Zhang[1], David Lum[2], Joseph E. Druso[1], Bryant Blank[3], Kristin F. Wilson[1], Alana Welm[2], Marc A. Antonyak[1] & Richard A. Cerione[1,4]

Non-classical secretory vesicles, collectively referred to as extracellular vesicles (EVs), have been implicated in different aspects of cancer cell survival and metastasis. Here, we describe how a specific class of EVs, called microvesicles (MVs), activates VEGF receptors and tumour angiogenesis through a unique 90 kDa form of VEGF (VEGF$_{90K}$). We show that VEGF$_{90K}$ is generated by the crosslinking of VEGF$_{165}$, catalysed by the enzyme tissue transglutaminase, and associates with MVs through its interaction with the chaperone Hsp90. We further demonstrate that MV-associated VEGF$_{90K}$ has a weakened affinity for Bevacizumab, causing Bevacizumab to be ineffective in blocking MV-dependent VEGF receptor activation. However, treatment with an Hsp90 inhibitor releases VEGF$_{90K}$ from MVs, restoring the sensitivity of VEGF$_{90K}$ to Bevacizumab. These findings reveal a novel mechanism by which cancer cell-derived MVs influence the tumour microenvironment and highlight the importance of recognizing their unique properties when considering drug treatment strategies.

[1] Department of Molecular Medicine, Cornell University, Ithaca, New York 14853, USA. [2] Huntsman Cancer Institute, University of Utah, Salt Lake City, Utah 84112, USA. [3] Department of Biomedical Sciences, Cornell University, Ithaca, New York 14853, USA. [4] Department of Chemistry and Chemical Biology, Cornell University, Ithaca, New York 14853, USA. † Present address: Eastern Hepatobiliary Surgery Institute, National Center for Liver Cancer, Shanghai 201805, P. R. China. Correspondence and requests for materials should be addressed to R.A.C. (email: rac1@cornell.edu).

Cell-to-cell communication plays fundamentally important roles in organismal development, tissue homeostasis, and when de-regulated, in various diseases including cancer. The importance of paracrine signalling in mediating such communication is well established, among the many examples being diffusible growth factors and pro-inflammatory cytokines that are secreted by one cell and then bind to their specific receptors expressed on the surfaces of nearby cells. However, the discovery of extracellular vesicles (EVs) represents an exciting area of research that offers novel mechanisms by which paracrine signalling can be achieved.

EVs are 'non-classical' secretory vesicles that fall into two broad classes based on their size; exosomes (typically < 100 nm in diameter), which are generated from endosomal re-cycling and multi-vesicular bodies, and larger vesicles, often referred to as microparticles or microvesicles (MVs), that range in size from 100 nm to more than 1 μm in diameter and are formed and shed from the plasma membrane[1–4]. The content of EVs is varied and includes growth factors, growth factor receptors, mRNAs and microRNAs; however, they also contain some selective cargo representative of their cells of origin[1,5]. When EVs are shed from their parental cancer cells, they are capable of transferring biomolecules and altering the signalling behaviour of neighbouring cells[6–8]. Thus, they provide important mechanisms by which cancer cells at a primary tumour site communicate with their immediate environment and have been implicated in a number of aspects of cancer progression and metastasis, including the creation of the pre-metastatic niche at secondary sites of tumour formation, as well as the promotion of tumour angiogenesis[5,7,9–15].

It has been reported that EVs contain VEGF[16], although the roles played by EV-associated VEGF are still not clear, and additional mechanisms have been proposed to explain how EVs might contribute to tumour angiogenesis[1]. Understanding how EVs stimulate this important process in cancer progression carries potentially significant consequences, given that blocking tumour angiogenesis has been a major anti-cancer strategy. A prime example has been the development of the monoclonal VEGF antibody Bevacizumab (trade name Avastin[17–19]), which has demonstrated some degree of efficacy in clinical trials when administered in combination with standard chemotherapy in non-small cell lung cancer, and as a first-line treatment in patients with metastatic renal cell carcinoma[20]. However, Bevaciuzmb has often failed to give complete responses, with the majority of patients developing resistance[18,20–25]. These findings highlight the need for a deeper understanding of how aggressive cancer cells promote tumour vascularization and colonize secondary tissues.

In this study, we set out to determine the underlying mechanisms by which the combination of Bevacizumab and an Hsp90 inhibitor gave rise to a striking synergistic inhibition of tumour growth and blood vessel formation in xenograft models for breast cancer. This led us to discover how a specific class of EVs (MVs) derived from breast cancer cells provides a sustained activation of VEGFRs on endothelial cells. We show that this involves an ∼ 90 kDa crosslinked form of VEGF (VEGF$_{90K}$) that associates with the outer surfaces of MVs. This unique form of VEGF is generated through the crosslinking of a smaller (165 amino acid) VEGF isoform, VEGF$_{165}$, by the acyl transferase tissue transglutaminase (tTG), and associates with MVs through its interaction with Hsp90. Moreover, we show that MV-associated VEGF$_{90K}$ has a reduced affinity for VEGF antibodies like Bevacizumab and is relatively insensitive to their inhibitory effects. However, treatment with Hsp90 inhibitors causes the release of VEGF$_{90K}$ from MVs, enabling it to bind to Bevacizumab, thus restoring its sensitivity to the drug. Taken together, these findings illustrate the roles played by cancer cell-derived MVs and their associated VEGF$_{90K}$ in the tumour microenvironment, and how they can contribute to tumour angiogenesis in a unique way that limits the effectiveness of Bevacizumab therapy.

## Results

**Combining Bevacizumab and 17AAG inhibits tumour growth**. During experiments aimed at identifying drug combinations that exhibit beneficial effects in blocking tumour growth in MDAMB231 cell xenografts in mice, we found that the combination of the VEGF drug Bevacizumab with an Hsp90 inhibitor gave striking results. Figure 1a shows the results of one such set of experiments. MDAMB231 cells were introduced into mice and then when the tumours reached a size ∼ 1–2 mm in diameter, the drug treatments were initiated. The combination of Bevacizumab with the Hsp90 inhibitor 17AAG (17-N-allylamino-17-demo-thoxygeldanamycin)[26] proved to be markedly more effective in blocking tumour growth in these mouse xenografts, compared with when either drug was administered alone.

We further examined the potential benefits of combining Bevacizumab with 17AAG in a breast cancer patient-derived mouse xenograft (PDX) model originally developed by DeRose et al.[27] (also described in Table 1). This breast cancer model offers some important advantages compared with the MDAMB231 xenograft system. In particular, the engraftment of patient-derived tumour tissues into immune-deficient mice has been suggested to provide a significant improvement over the subcutaneous implantation of cancer cell lines in pheno-copying human tumours and in predicting drug responses in patients[28–35].

Similar to what we observed in MDAMB231 cell xenografts, treatment of mice containing either the triple-negative tumour graft HCI-001 or HCI-002 with both Bevacizumab and 17AAG resulted in a marked improvement by strongly blocking tumour growth, compared with when treating with each individual drug (Fig. 1b,c). The combination of 17AAG and Bevacizumab also strongly inhibited tumour angiogenesis in these animals and induced hypoxia (Supplementary Fig. 1A,B). Given these findings, we set out to establish the mechanistic basis by which combining Bevacizumab with an Hsp90 inhibitor was able to provide such beneficial effects in MDAMB231 xenografts and the triple-negative PDXs.

**Reconstituting the effects of Bevacizumab and 17AAG**. As a first step towards delineating the molecular mechanism that underlies the synergy observed when using Bevacizumab and an Hsp90 inhibitor to block tumour growth in mice, we sought to reconstitute these effects in well defined models for angiogenesis. We began by establishing a system to assay the ability of MDAMB231 cells to activate endothelial cells and stimulate their migration into small capsules (called 'angioreactors') implanted into mice (Fig. 2a)[28]. MDAMB231 cells were able to effectively stimulate the migration of endothelial cells into angioreactors, with their stimulatory activity being relatively insensitive to 17AAG and less sensitive to inactivating VEGF antibodies (that is, when pre-incubated with the cells in the capsules), compared with the stimulatory activity of the recombinant, 165 amino acid VEGF molecule (rVEGF$_{165}$) that represents one of the predominant isoforms of VEGF (Fig. 2b,c; Supplementary Fig. 2A).

We then set out to probe the mechanism by which MDAMB231 cells exhibited reduced sensitivity to Bevacizumab. As a starting point, we examined the potential role of EVs,

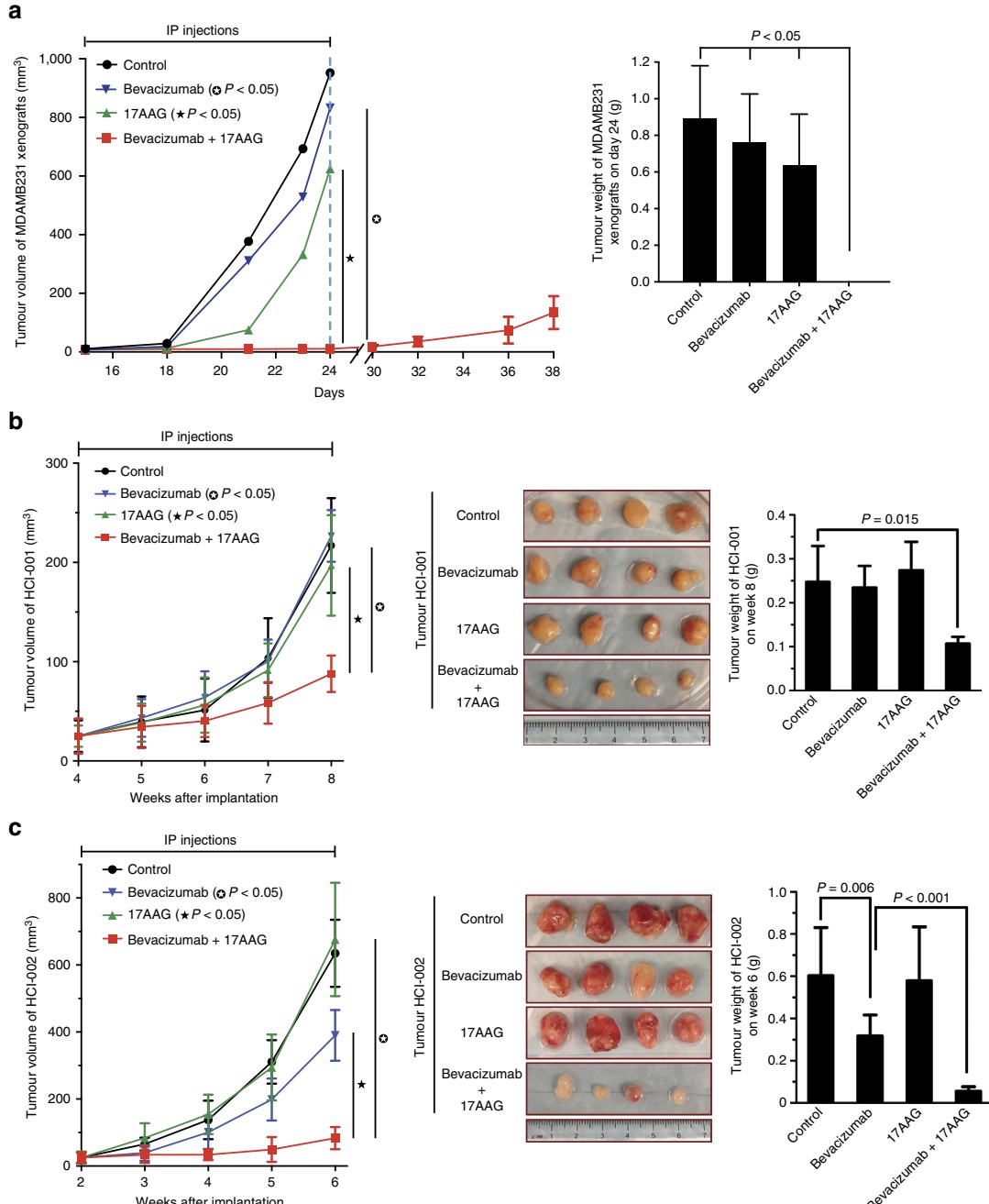

**Figure 1 | Combining an Hsp90 inhibitor and Bevacizumab synergistically inhibits tumour growth. (a)** MDAMB231 cells ($3 \times 10^6$) were injected subcutaneously into each flank of nude mice. On detecting tumours of 1–2 mm in diameter, IP injections of vehicle only (control), Bevacizumab (5 mg kg$^{-1}$), 17AAG (20 mg kg$^{-1}$) or Bevacizumab plus 17AAG, were initiated and the drugs were re-administered every third day for 15 days. Left: Plots showing mean tumour volumes (mm$^3$) as a function of time for mice treated with the various drug combinations ($n = 4$ for each condition). The tumour volumes for the Bevacizumab plus 17AAG treatment group, versus treatment with Bevacizumab or 17AAG alone, were statistically significant ($P < 0.05$). Right: Histograms showing mean weights of the resulting MDAMB231 xenografts 24 days after being treated with different drug combinations. **(b)** Human breast cancer HCI-001 tumour grafts[27] were implanted into the fat pads of NOD/SCID mice. When the tumours reached 3 mm in diameter (~4 weeks after implantation), IP injections with vehicle only (control), Bevacizumab (2.5 mg kg$^{-1}$), 17AAG (10 mg kg$^{-1}$) or Bevacizumab plus 17AAG, were initiated every other day for 4 weeks. Left: Plots showing mean tumour volumes (mm$^3$) as a function of time for mice treated with the various drug combinations ($n = 4$ for each condition). The tumour volumes for the Bevacizumab plus 17AAG treatment group, versus treatment with Bevacizumab or 17AAG alone, were statistically significant ($P < 0.05$). Middle and Right: Images and histograms showing the resulting HCI-001 tumours, and mean weights of these tumours, 8 weeks after being implanted and treated with the different drug combinations. **(c)** Left: Plots showing the mean tumour volumes (mm$^3$) of tumour graft HCI-002 grown in mice treated with vehicle only (control), Bevacizumab, 17AAG or Bevacizumab plus 17AAG, at the indicated intervals ($n = 8$ for each condition). The tumour volumes for the Bevacizumab plus 17AAG treatment group, versus the Bevacizumab treatment group or 17AAG treatment group, were statistically significant ($P < 0.05$). Middle and Right: Images and histograms showing the HCI-002 tumours, and mean weights of these tumours, 6 weeks after being implanted and treated with the different drug combinations.

**Table 1 | Tumour samples.**

| ID | Source | Primary diagnosis | ER, PR, HER2 status |
|---|---|---|---|
| HCI-001 | Breast tumour | IDC | −/−/− |
| HCI-002 | Breast tumour | IDC | −/−/− |
| HCI-003 | Breast tumour | IDC | +/+/− |
| HCI-005 | Pleural effusion | IDC and ILC | +/+/+ |
| HCI-006 | Pleural effusion | — | +/+/+ |
| HCI-007 | Pleural effusion | IBC | +/+/+ |
| HCI-008 | Pleural effusion | Poorly differentiated adenocarcinoma | −/−/+ |
| HCI-009 | Ascites | Poorly differentiated adenocarcinoma | −/−/− |
| HCI-010 | Pleural effusion | IDC | −/−/− |
| HCI-011 | Pleural effusion | IDC | +/+/− |
| HCI-012 | Pleural effusion | IDC | −/−/+ |
| HCI-013 | Breast tumour | — | +/+/− |

IBC, inflammatory breast cancer; IDC, infiltrating ductal carcinoma; ILC, infiltrating lobular carcinoma.
Table describing the names (IDs) of 12 human tumour samples, their source, primary diagnosis, oestrogen receptor (ER), progesterone receptor (PR) and HER2 status (also see DeRose et al.[27]).

given their suggested involvement in different stages of cancer progression, including their ability to stimulate tumour angiogenesis[5–7,10–12]. We were particularly interested in examining a group of larger sized, actin filament-based EVs, that we will refer to from here on as MVs. MDAMB231 cells are much more prolific than their non-transformed cellular counterparts in generating large MVs[7,11], which can be visualized by electron microscopy (Fig. 2d, arrows), as well as by rhodamine-conjugated phalloidin staining of F-actin[7] (also, see below). They can be isolated from the conditioned medium of MDAMB231 cells by a series of steps (Supplementary Fig. 2B) that begin with either consecutive centrifugations ($\sim 300g$), or filtration on a 3.1 μm filter, to remove larger cellular debris, followed by filtration on a 0.1 or 0.22 μm filter (that is, the MVs are retained on these filters). The initial steps of low speed centrifugation, or filtration of the conditioned medium on a 3.1 μm filter, are necessary to ensure the capture of sufficient amounts of the larger, actin-associated MVs by the subsequent filtration step, because a significant fraction of those vesicles are lost (pelleted) when higher sedimentation forces are used to remove cellular debris. These procedures yielded MVs ranging in size from 0.5–1 μm in diameter, as assessed by fluorescent staining with the membrane dye FM 1-43FX (Fig. 2e) or rhodamine-conjugated phalloidin (Fig. 2f).

Preparations of MVs isolated from MDAMB231 cells strongly stimulated endothelial cell migration into the angioreactors (Fig. 2g), as did MVs prepared from another human breast cancer cell line (SKBR3 adenocarcinoma cells), whereas MVs from human cervical carcinoma (HeLa) cells were ineffective (Supplementary Fig. 2C). As was the case when assaying the tumour growth of MDAMB231 xenografts, the ability of MVs isolated from breast cancer cells to stimulate endothelial cell migration into angioreactors was relatively insensitive to Bevacizumab. However, when MV preparations from MDAMB231 cells were treated with 17AAG, their ability to stimulate this response was highly sensitive to Bevacizumab (Fig. 2h), much like we observed when combining 17AAG and Bevacizumab in MDAMB231 xenografts and the triple-negative PDXs (Fig. 1a–c).

Similar results were obtained when using an *in vitro* assay for angiogenesis that monitored the ability of HUVECs to undergo tubulogenesis[36]. MVs prepared from MDAMB231 were capable of stimulating HUVECs to undergo tubulogenesis (Fig. 2i). Pre-incubating rVEGF$_{165}$ with Bevacizumab, before its addition to HUVECs, strongly inhibited its ability to stimulate tubulogenesis, whereas as a control, treatment with 17AAG was without effect (Supplementary Fig. 2D). While pre-incubating

MDAMB231 cell-derived MVs with the anti-VEGF antibody alone failed to block their stimulatory capability; when MVs were incubated with the combination of Bevacizumab and 17AAG, their ability to stimulate tubulogenesis was strongly inhibited (Fig. 2j).

**Identification of a unique form of VEGF in breast cancer MVs.** The results presented in Fig. 2 suggested a fundamental difference between the abilities of soluble recombinant VEGF and MV-associated VEGF to be recognized by VEGF antibodies. Cancer cells produce several splice variants of VEGF (also known as VEGF-A) that range in size from $M_r \sim 12–25$ kDa[16,17,19,37,38]. Analysis of MDAMB231 cells by RT-PCR, using primer sets that amplified the full-length VEGF-A transcript or individual VEGF-A isoforms, detected the following splice variants of VEGF-A; VEGF$_{121}$, VEGF$_{165}$ and VEGF$_{189}$ (the subscripts denote the number of amino acids for each species (Supplementary Fig. 3A,B)). In addition, larger forms of VEGF ($M_r$ 70–110 kDa) were detected when analysing cell lysates derived from MDAMB231 cells by western blotting with a pan VEGF antibody (Supplementary Fig. 3C), with a 90 kDa species (VEGF$_{90K}$) being detected when using a pan VEGF antibody or an antibody that specifically recognizes VEGF$_{165}$ (Fig. 3a). Analysis of this 90 kDa band by mass spectrometry confirmed that it contained sequences matching VEGF-A.

MDAMB231 cells secreted VEGF$_{90K}$ into the medium, as well as a second smaller VEGF species (Fig. 3b). The same two VEGF species were detected in the conditioned medium of SKBR3 breast cancer cells, whereas, VEGF$_{90K}$ was absent from the medium collected from human cervical carcinoma HeLa cells (Fig. 3b).

MVs can be visualized along the surfaces of cancer cells by immunofluorescence when staining actin with rhodamine-conjugated phalloidin, and when using the lipid-binding dye, FM1-43X (Supplementary Fig. 3D). Immunofluorescence experiments using either a pan VEGF antibody or an anti-VEGF$_{165}$ antibody showed that VEGF is present together with actin and the lipid microdomain protein flotillin-2 in MVs along the surfaces of permeabilized MDAMB231 cells (Fig. 3c; Supplementary Fig. 3E). When we isolated the two major classes of EVs from the conditioned medium, we found that only MVs contained VEGF$_{90K}$ and actin, when western blotting vesicle lysates with a pan VEGF antibody and an anti-actin antibody, but lacked the smaller molecular mass VEGF species. Neither VEGF$_{90K}$ nor any other form of VEGF was

detected in exosomes, that is, EVs that pass through the 0.22 μm filter and contain the exosome marker protein CD-63 (ref. 39), and both classes of EVs contained flotillin-2 (Fig. 3d). Likewise, we detected only $VEGF_{90K}$ in MVs isolated from MDAMB231 and SKBR3 cells in western blots using the antibody that specifically recognizes $VEGF_{165}$ (Fig. 3e). Although MVs shed by the human cervical carcinoma HeLa cell line did not have any detectable VEGF (Fig. 3e; Supplementary Fig. 3C); interestingly, we detected MVs with associated $VEGF_{90K}$ in other human cancer cell lines including

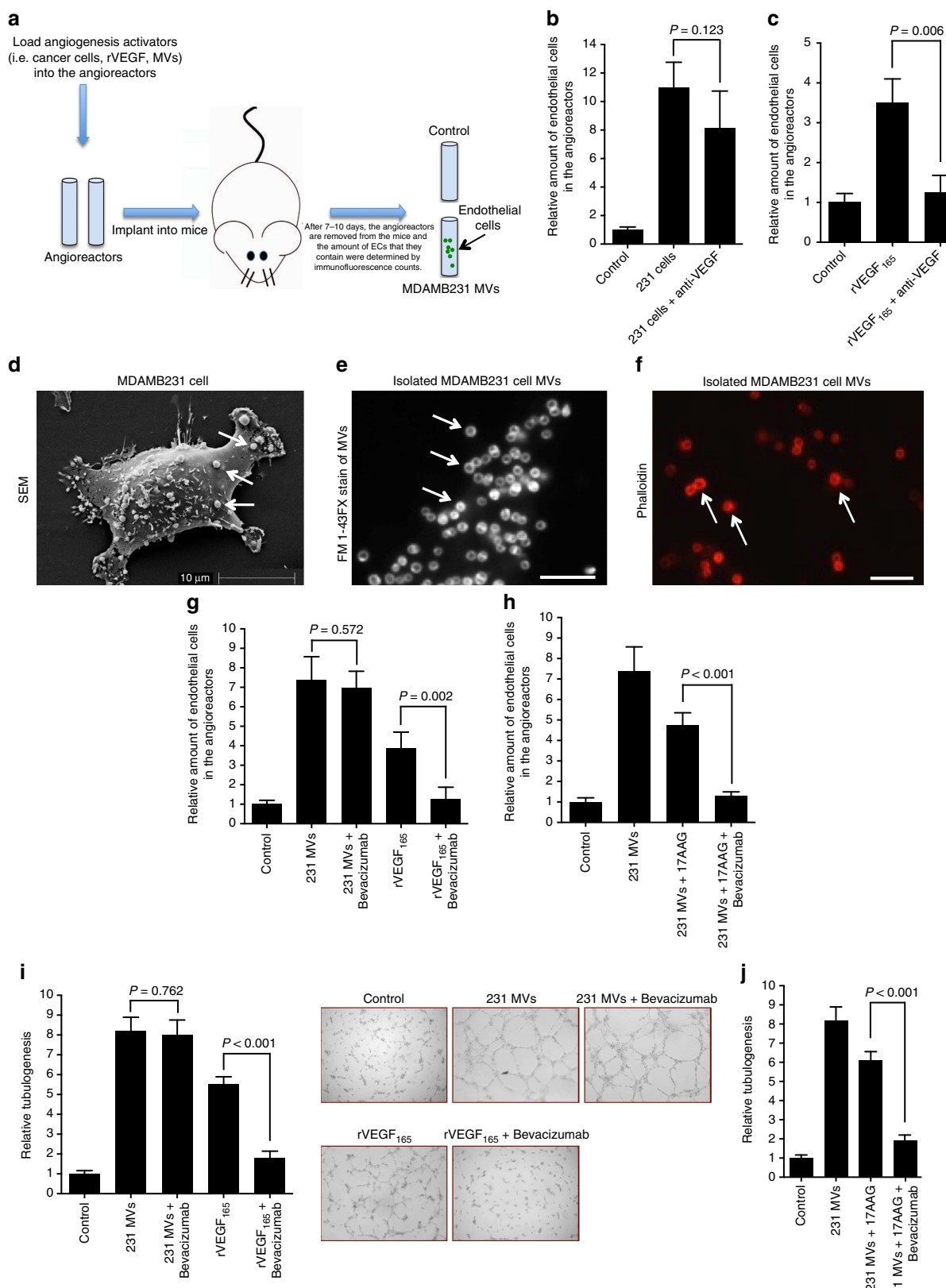

U87 glioblastoma cells and HT29 colorectal adenocarcinoma cells (Supplementary Fig. 3F).

MDAMB231 cells transfected with siRNAs targeting all forms of VEGF (Fig. 3f, second panel from the bottom) showed little detectable $VEGF_{90K}$ in their MVs, compared with MVs prepared from cells expressing control siRNA (Fig. 3f, top panel). Consequently, only MVs prepared from cells treated with the control RNAi stimulated HUVECs to undergo tubulogenesis (Fig. 3g), demonstrating that MV-associated VEGF is essential for stimulating this process.

**Crosslinked $VEGF_{90K}$ is formed by tissue transglutaminase.** While a 90 kDa VEGF species had been previously observed, its origin was unknown[37,38]. $VEGF_{90K}$ is not a splice variant of VEGF, however, it can be detected using an antibody that specifically recognizes $VEGF_{165}$, raising the possibility that $VEGF_{90K}$ is a larger crosslinked form of $VEGF_{165}$. An intriguing candidate for catalysing the formation of $VEGF_{90K}$ was the acyl transferase tissue transglutaminase (tTG)[40], since this protein crosslinking enzyme is over-expressed in MDAMB231 cells as well as in other highly aggressive human breast carcinomas[41,42]. Moreover, tTG is secreted from cells and associated with cancer cell-derived MVs, where its protein crosslinking catalytic site is accessible along their outer surface[7]. Indeed, we found that $rVEGF_{165}$ can be crosslinked to form $VEGF_{90K}$ *in vitro* (see below), and that RNAi-mediated knockdowns of tTG in MDAMB231 cells (Fig. 3h, fourth panel from top) significantly reduced the amount of $VEGF_{90K}$ in MVs (Fig. 3h, top panel). The same was true when treating the cells with the tTG crosslinking inhibitor monodansylcadaverine (MDC). These treatments did not affect flotillin-2 levels in the MVs (Fig. 3h, second panel from the top) and therefore did not cause a general elimination of all MV-associated proteins. Accordingly, MVs prepared from MDAMB231 cells in which tTG had been knocked-down, or its catalytic activity disabled by treatment with MDC, were less effective at stimulating HUVEC tubulogenesis (Fig. 3i).

**MVs activate VEGFRs in a Bevacizumab-insensitive manner.** We next set out to determine what functional distinctions might exist between MV-associated VEGF and soluble VEGF. When the ability of MDAMB231 cell-derived MVs to stimulate HUVEC tubulogenesis was compared with that of $rVEGF_{165}$, under conditions where the levels of MV-associated $VEGF_{90K}$ were matched to those of $rVEGF_{165}$ by western blotting using $rVEGF_{165}$ as a standard (Supplementary Fig. 4A), the MVs exhibited an ~2-fold greater stimulatory capability (Fig. 4a). MVs collected from SKBR3 cells performed similarly,

whereas MVs from HeLa cells that lack $VEGF_{90K}$ only weakly stimulated tubulogenesis.

We then examined the ability of breast cancer cell-derived MVs to activate VEGFRs on HUVECs, where VEGFR2 is predominantly responsible for stimulating angiogenesis[17,19,43]. The addition of $rVEGF_{165}$ to HUVECs caused a transient activation of both VEGFR2 and its downstream signalling target ERK, which peaked within ~15 min and was significantly diminished after 45 min (Fig. 4b, top two panels, lanes 1–3), whereas MDAMB231 cell-derived MVs caused a more sustained stimulation of VEGFR2-signalling and ERK activation (Fig. 4b, top two panels, lanes 4–6) that continued through 90 min (Supplementary Fig. 4B). MVs from MDAMB231 cells, in which VEGF expression was first knocked-down by siRNA, showed little ability to stimulate VEGFR2-signalling activities (Fig. 4c).

The activation of VEGFRs by MDAMB231 cell-derived MVs was unaffected by either Bevacizumab or a pan anti-VEGF antibody (Fig. 4d, lanes 2–4). Similarly, the ability of MVs from SKBR3 breast cancer cells to stimulate VEGFR2 activation was only minimally affected by Bevacizumab (Fig. 4e). This was in sharp contrast to $rVEGF_{165}$, whose stimulatory activity was blocked by the antibody (Fig. 4d, lanes 5 and 6). These findings were consistent with the results from the tubulogenesis assays and angioreactor experiments described in Fig. 2g–j, and suggested that VEGF antibodies bind to MV-associated VEGF less effectively than to free VEGF. This was further supported by our finding that $VEGF_{90K}$, when associated with intact MVs, was not able to be immunoprecipitated with a VEGF antibody, whereas it could be immunoprecipitated when released from MVs on their lysis by detergents (Fig. 4f).

**Hsp90 localizes to MVs and binds $VEGF_{90K}$.** Having established that MV-associated $VEGF_{90K}$ has properties distinct from soluble VEGF, we investigated whether a relationship between MV-associated $VEGF_{90K}$ and Hsp90 might underlie the effectiveness of Bevacizumab/Hsp90 co-therapy observed in Fig. 1a–c. Hsp90 is a molecular chaperone that promotes protein-folding, intracellular transport and signalling events[44–46], and has been shown to be associated with EVs[7,47,48]. VEGF is a client protein of Hsp90 (ref. 49), and immunofluorescence experiments show that Hsp90 localizes with VEGF in MVs generated by various breast cancer cells, including primary tumour cells cultured from PDXs (Fig. 5a, see arrows), as well as MDAMB231 cells (see Fig. 6a, below), but not in cells in which VEGF expression was eliminated by RNAi (Supplementary Fig. 5A). A movie showing the 3D rotation of a MV derived from MDAMB231 cells (Supplementary Movie 1) demonstrates

**Figure 2 | Reconstitution of Bevacizumab and 17AAG sensitivity.** (**a**) Schematic of the *in vivo* angiogenesis assay. (**b,c**) Relative amounts of endothelial cells that entered implanted angioreactors that lacked any activators (vehicle control; histogram 1 in both graphs), or were loaded with MDAMB231 cells ($5 \times 10^4$ cells per angioreactor) without (**b**, histogram 2) or with a pan inactivating VEGF antibody (200 ng per angioreactor; **b** histogram 3) or were loaded with $rVEGF_{165}$ (4 ng per angioreactor) without (**c** histogram 2) or with VEGF antibody (200 ng per angioreactor; C, histogram 3). (**d**) Scanning electron microscopy (s.e.m.) of a MDAMB231 cell. Arrows indicate large EVs. (**e**) MVs from MDAMB231 cells were examined by fluorescence staining using the membrane dye FM 1-43FX. Arrows indicate MVs. Scale bar, 5 μm. (**f**) MVs isolated from MDAMB231 cells were examined using rhodamine-conjugated phalloidin. Arrows indicate MVs. Scale bar, 5 μm. (**g**) The relative amounts of endothelial cells that entered angioreactors that lacked activators (control; histogram 1), contained MDAMB231 cell MVs (2 μg total protein per angioreactor) without (histogram 2) or with Bevacizumab (1 μg per angioreactor; histogram 3) or contained $rVEGF_{165}$ (4 ng per angioreactor) without (histogram 4) or with Bevacizumab (1 μg per angioreactor; histogram 5). (**h**) Relative amounts of endothelial cells that entered angioreactors that lacked activators (vehicle control; histogram 1), or angioreactors that contained the indicated combinations of MDAMB231 cell MVs (2 μg per angioreactor), 10 μM 17AAG and 1 μg Bevacizumab (histograms 2-4). (**i**) Tubulogenesis assays of HUVECs left untreated (control; histogram 1), treated with MVs (10 μg ml$^{-1}$ total protein) from MDAMB231 cells without (histogram 2) or with 0.5 μg ml$^{-1}$ Bevacizumab (histogram 3) or treated with rVEGF (15 ng ml$^{-1}$) without (histogram 4) or with 0.5 μg ml$^{-1}$ Bevacizumab (histogram 5). Left: The relative differences in tube lengths were plotted. Right: Images of the tubulogenesis assays performed under the different conditions tested. (**j**) Tubulogenesis assays were performed on HUVECs that were untreated (control; histogram 1) or treated with MVs (10 μg ml$^{-1}$ of MV protein) from MDAMB231 cells pre-treated with various combinations of 10 μM 17AAG and 0.5 μg ml$^{-1}$ Bevacizumab (histograms 2-5).

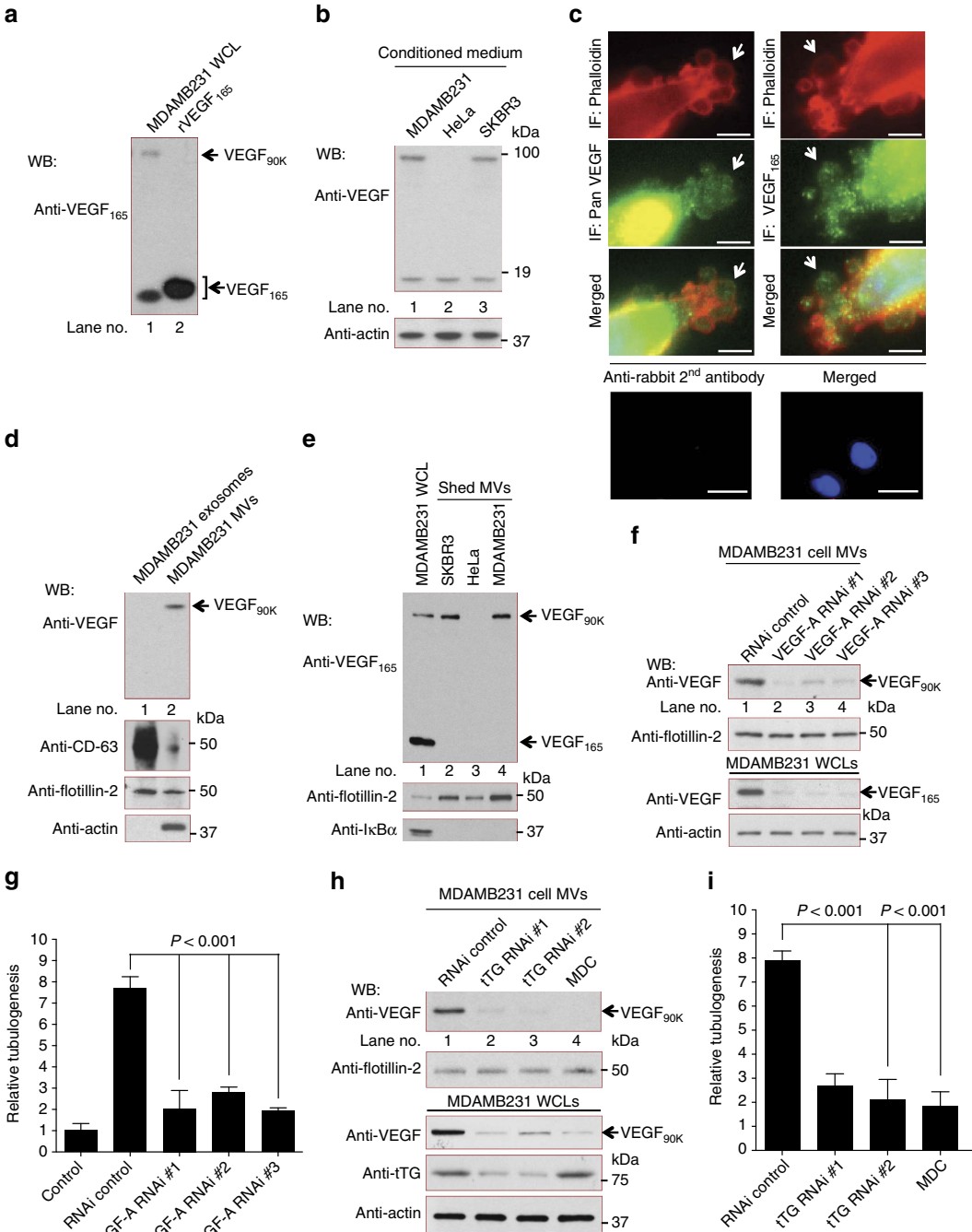

**Figure 3 | MVs shed from breast cancer cells contain an oligomeric VEGF species.** (**a**) Whole cell lysates (WCL) from MDAMB231 cells (lane 1) and human recombinant VEGF$_{165}$ (rVEGF$_{165}$; lane 2) were immunoblotted with antibodies against VEGF$_{165}$. (**b**) Concentrated conditioned medium (20 μg total protein) from serum-starved MDAMB231 (lane 1), HeLa (lane 2) or SKBR3 cells (lane 3) were immunoblotted. (**c**) MDAMB231 cells were analysed by immunofluorescent (IF) microscopy using Rhodamine-conjugated phalloidin (top panels) and either a pan VEGF or anti-VEGF$_{165}$ antibody (middle panels). Arrows indicate VEGF localized on MVs. Scale bar, 2 μm. IF images of MDAMB231 cells stained by secondary antibody (control; bottom panels). Scale bar, 10 μm (**d**) Exosomes (lane 1) or MVs (lane 2) from MDAMB231 cells were isolated, lysed and immunoblotted (5 μg per samples) with antibodies against VEGF, CD-63, actin and flotillin-2. (**e**) MVs from SKBR3 (lane 2), HeLa (lane 3) or MDAMB231 (lane 4) cells were isolated and lysed. WCL from MDAMB231 cells (lane 1), as well as MV lysates (10 μg per sample), were immunoblotted with antibodies against VEGF$_{165}$, flotillin-2 and the cytosolic-specific marker IκBα. (**f**) MVs from MDAMB231 cells transfected with control siRNA (lane 1) or siRNAs targeting VEGF (lanes 2–4) were immunoblotted with a pan VEGF antibody or anti-flotillin-2. WCL were immunoblotted with a pan VEGF antibody and an anti-actin antibody. (**g**) Tubulogenesis assays on HUVECs that were untreated (control) or treated with MVs from MDAMB231 cells (10 μg ml$^{-1}$ MV protein) transfected with control siRNA or siRNAs targeting VEGF. (**h**) MVs from MDAMB231 cells transfected with control siRNA (lane 1) or siRNAs targeting tTG (lanes 2 and 3) or treated with MDC (50 μM) (lane 4) were immunoblotted with antibodies against VEGF or flotillin-2, while WCLs were immunoblotted with antibodies against VEGF, tTG or actin. (**i**) Tubulogenesis assays of HUVECs treated with MVs (10 μg ml$^{-1}$ MV protein) from MDAMB231 cells expressing control siRNA, siRNAs targeting tTG or with MVs from MDAMB231 cells treated with 50 μM MDC.

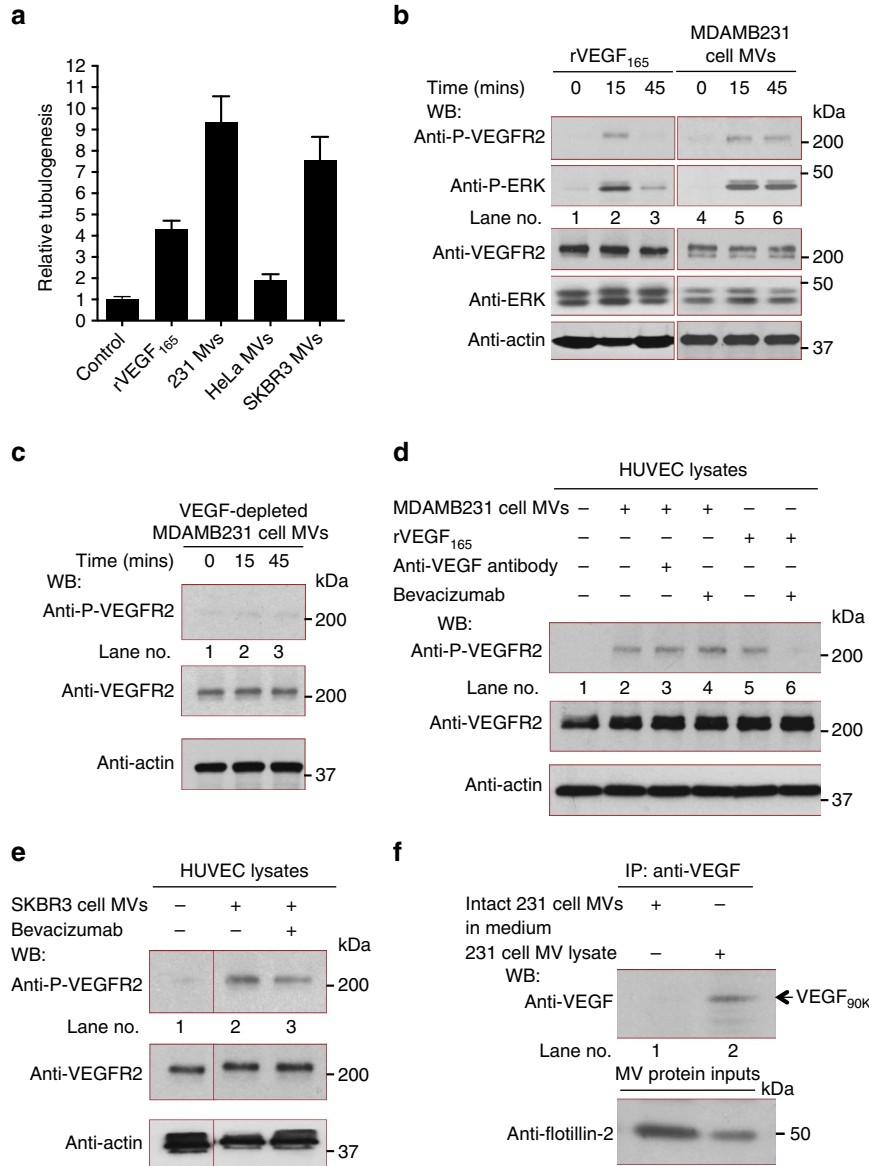

**Figure 4 | MV-associated VEGF$_{90K}$ stimulates a sustained activation of VEGFRs that is insensitive to Bevacizumab. (a)** Tubulogenesis assays were performed on HUVECs that were untreated (control; histogram 1) or treated with recombinant VEGF$_{165}$ (rVEGF; 15 ng ml$^{-1}$) (histogram 2) or with MVs (10 μg ml$^{-1}$ total protein) from MDAMB231 (histogram 3), HeLa (histogram 4) or SKBR3 (histogram 5) cells. **(b)** Lysates of serum-deprived HUVECs exposed to rVEGF$_{165}$ (5 ng ml$^{-1}$; lanes 1–3) or MVs from MDAMB231 cells (5 μg ml$^{-1}$ of MV protein; lanes 4–6) for the indicated lengths of time were immunoblotted with antibodies that recognize phosphorylated VEGFR2 (P-VEGFR2), total VEGFR2, phosphorylated ERK (P-ERK), total ERK or actin. **(c)** Lysates of serum-deprived HUVECs treated for the indicated time with MVs (5 μg ml$^{-1}$ of MV protein) from MDAMB231 cells expressing VEGF siRNA were immunoblotted with antibodies that recognize phosphorylated VEGFR2, total VEGFR2 or actin. **(d)** Lysates of serum-deprived cultures of HUVECs that were untreated (lane 1), treated with MDAMB231 cell MVs (5 μg ml$^{-1}$ of MV protein) without (lane 2) or with (lane 3) either 200 ng ml$^{-1}$ anti-pan VEGF antibody, 0.5 μg ml$^{-1}$ Bevacizumab (lane 4) or rVEGF$_{165}$ (5 ng ml$^{-1}$) without (lane 5) or with (lane 6) 0.5 μg ml$^{-1}$ Bevacizumab, for 15 min were immunoblotted with antibodies that recognize phosphorylated VEGFR2, total VEGFR2 or actin. **(e)** Lysates of serum-deprived HUVECs that were untreated (lane 1), or exposed to SKBR3 cell MVs (5 μg ml$^{-1}$ total protein) treated without (lane 2) or with (lane 3) 0.5 μg ml$^{-1}$ Bevacizumab for 15 min were immunoblotted with antibodies that recognize phosphorylated VEGFR2, total VEGFR2 or actin. **(f)** MDAMB231 cell MVs were evenly divided into two samples. In one sample, immunoprecipitations (IPs) using a pan VEGF antibody were performed on the intact MVs (∼25 μg of MV protein in RPMI medium; lane 1), while in the other sample, MVs were first lysed before immunoprecipitations were performed (lane 2). The immunocomplexes were blotted with a pan VEGF antibody and the MV protein inputs were blotted with an antibody against the MV marker flotillin-2.

that Hsp90 and VEGF are located along the vesicle surface. Moreover, Hsp90 can be co-immunoprecipitated with VEGF$_{90K}$ after lysing MVs prepared from MDAMB231 cells (Fig. 5b, lane 3). HeLa cell MVs that lack VEGF$_{90K}$ (Fig. 5b, lane 2), and MDAMB231 cell MVs that were not incubated with a pan VEGF antibody (Fig. 5b, lane 1), served as negative controls.

We generated VEGF$_{90K}$ by incubating rVEGF$_{165}$ with purified tTG under conditions optimal for tTG-catalysed protein crosslinking and found that the newly generated VEGF$_{90K}$ could be co-immunoprecipitated with recombinant Hsp90 (Fig. 5c, top panel, lane 3), demonstrating that Hsp90 directly associates with VEGF$_{90K}$. Supplementary Fig. 5B (lane 2) shows

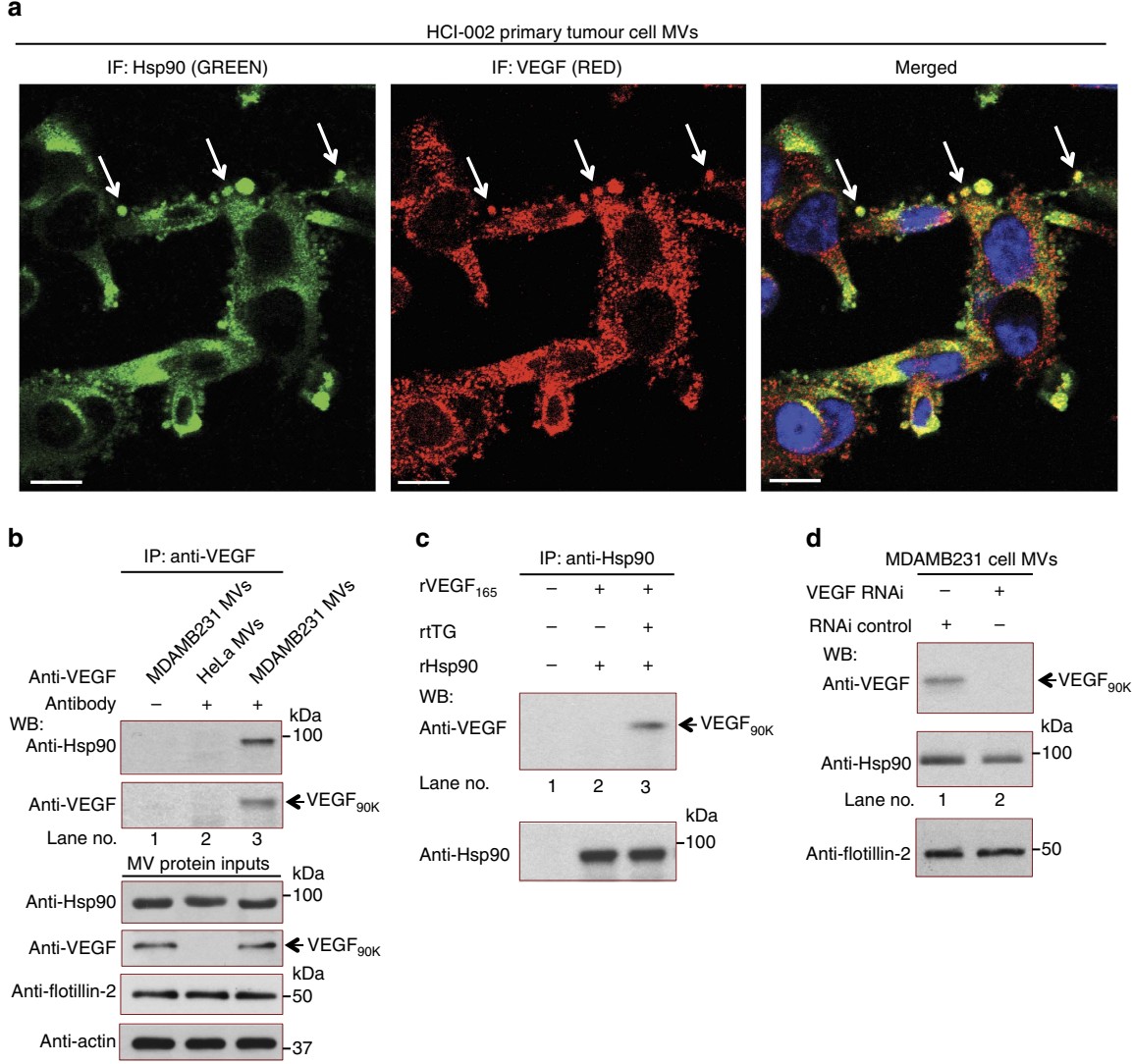

**Figure 5 | Hsp90 localizes to MVs and binds VEGF$_{90K}$.** (**a**) Immunofluorescent (IF) images of primary tumour cells cultured from breast cancer patient-derived xenograft HCI-002 stained with antibodies against Hsp90 and VEGF. Arrows indicate Hsp90 and VEGF localized on MVs. Scale bar, 10 μm. (**b**) MVs from serum-starved MDAMB231 (lanes 1 and 3) or HeLa (lane 2) cells were isolated and lysed. Immunoprecipitations (IPs) using a pan VEGF antibody were performed on the MV lysates (25 μg MV protein, each; lanes 2 and 3). As a control, MV lysates were incubated without the VEGF antibody (lane 1). The immunocomplexes were blotted with VEGF and Hsp90 antibodies. MV lysates were probed with antibodies against Hsp90, VEGF, flotillin-2 and actin (to confirm equivalent amounts of sample were used in each IP) (bottom panel). (**c**) rVEGF$_{165}$ (30 ng) was incubated in RPMI medium with recombinant Hsp90 (rHsp90; 30 ng, lanes 2 and 3), and without (lane 2) or with recombinant tTG (100 ng, lane 3), for 1h on ice to generate VEGF$_{90K}$. Immunoprecipitations (IPs) using an Hsp90 antibody were performed on the protein incubations and the immunocomplexes were blotted with antibodies against VEGF and Hsp90. As a control, rVEGF$_{165}$ was incubated without VEGF antibody (lane 1). (**d**) Lysates of MVs from MDAMB231 cells expressing control siRNA (lanes 1) or VEGF siRNA (lane 2) were immunoblotted with antibodies against VEGF, Hsp90 and the MV marker flotillin-2.

the corresponding control that VEGF$_{165}$, when incubated with tTG under crosslinking conditions in the absence of recombinant Hsp90, was unable to be immunoprecipitated with an anti-Hsp90 antibody. MVs prepared from MDAMB231 cells expressing control or VEGF-targeted siRNAs contained equivalent amounts of Hsp90 (Fig. 5d, middle panel), indicating that the interaction of VEGF$_{90K}$ with Hsp90 is not essential for localizing Hsp90 to the vesicles. Hsp90 has been identified on the cell surface[47], suggesting that it can directly associate with maturing MVs, independent of its association with VEGF$_{90K}$.

**VEGF released from MVs regains sensitivity to Bevacizumab.**
We hypothesized that since Hsp90 binds and localizes VEGF$_{90K}$

to the surfaces of MVs, then Hsp90 inhibitors that disrupt chaperone-client interactions should trigger the release of VEGF$_{90K}$ from the MVs, and thereby restore Bevacizumab sensitivity. MDAMB231 cells treated with the Hsp90 inhibitor 17AAG yielded MVs that still contained Hsp90 but were devoid of VEGF (Fig. 6a, see the arrows pointing to the MVs in the top and bottom panels; Supplementary Fig. 6 shows the secondary antibody controls). The loss of VEGF-staining from the MVs on 17AAG treatment suggested that VEGF$_{90K}$ had dissociated from Hsp90. Indeed, after treating MVs prepared from MDAMB231 cells with 17AAG, VEGF$_{90K}$ was not able to be co-immunoprecipitated with Hsp90 from the lysed vesicles (Fig. 6b). Similarly, when recombinant VEGF$_{90K}$ was generated by the tTG-catalysed crosslinking of rVEGF$_{165}$,

and then incubated with recombinant Hsp90 to form a $VEGF_{90K}$–Hsp90 complex *in vitro*, as demonstrated by the co-immunoprecipitation of these proteins, the addition of 17AAG caused Hsp90 to dissociate from $VEGF_{90K}$ (Fig. 6c).

When MVs isolated from MDAMB231 cells were treated with 17AAG and then collected again on 0.22 μm filters, the flow-through from the filtration contained the majority of the $VEGF_{90K}$ due to its dissociation from the MVs (Fig. 6d, compare

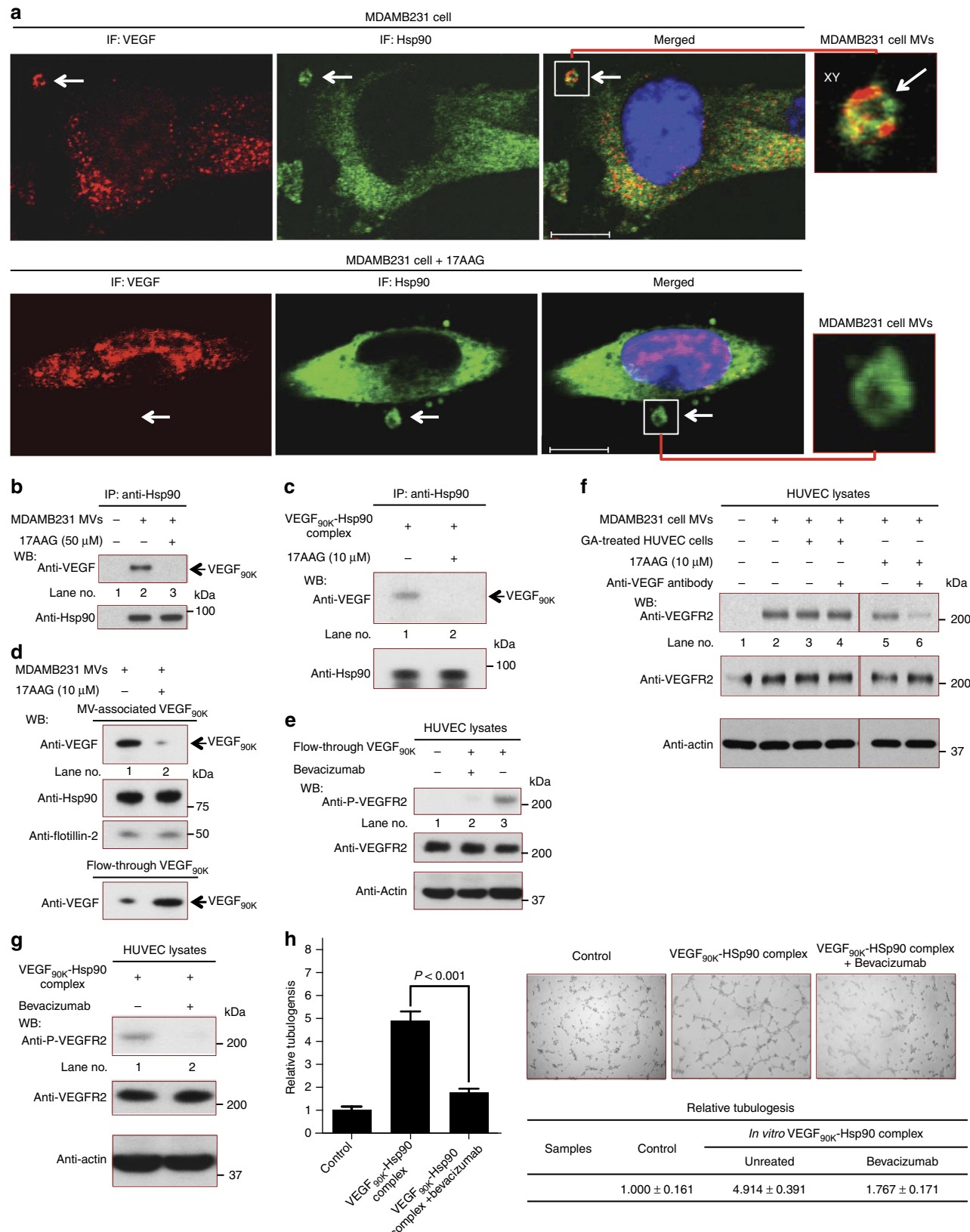

the top and bottom panels). $VEGF_{90K}$ from the flow-through was still capable of stimulating VEGFR2 auto-phosphorylation, although not in the presence of Bevacizumab (Fig. 6e). Similarly, $VEGF_{90K}$ that was released from MVs on their treatment with 17AAG and then directly assayed (that is, without filtration) still activated VEGFRs (Fig. 6f, compare lane 5 versus lane 2); however, its sensitivity to VEGF antibodies was restored (Fig. 6f, compare lanes 5 and 6). Because it was earlier reported that endogenous Hsp90 within endothelial cells can promote crosstalk between VEGFRs and integrins to help mediate VEGF-signalling[50], we wanted to verify that there was no contribution from endogenous Hsp90 to the responses of HUVECs to MV-associated $VEGF_{90K}$. In fact, we found that the addition of MVs to HUVECs that were pre-treated with a high affinity Hsp90 inhibitor, geldanamycin (GA), to eliminate any contributions from the endogenous Hsp90 within the recipient cells, still showed a stimulation of VEGFR2 auto-phosphorylation which was resistant to VEGF antibodies (Fig. 6f, compare lanes 2, 3 and 4).

We then further verified that the insensitivity to Bevacizumab exhibited by $VEGF_{90K}$ was not simply due to its interaction with Hsp90 (that is, such that it would occur even in the absence of MVs). Specifically, recombinant $VEGF_{90K}$ was formed by the tTG-catalysed crosslinking of $rVEGF_{165}$, and then incubated with recombinant Hsp90. The resultant $VEGF_{90K}$–Hsp90 complex-activated VEGFRs on HUVECs (Fig. 6g) and stimulated tubulogenesis (Fig. 6h); however, these stimulatory events were fully blocked by Bevacizumab. Thus taken together, these findings demonstrate that the insensitivity of $VEGF_{90K}$ to Bevacizumab only occurs when $VEGF_{90K}$ is associated (via Hsp90) with MVs.

**Insensitivity of MV-VEGF to Bevacizumab extends to PDXs.** We were interested in seeing whether the results we obtained with MV-associated $VEGF_{90K}$ in MDAMB231 cells would extend to experiments performed on a panel of breast cancer-derived PDXs (Table 1)[27]. Indeed, we found that the same VEGF species observed in the conditioned medium from MDAMB231 cells were present in the conditioned medium from a number of the PDX samples (Supplementary Fig. 7A; Supplementary Fig. 7B shows the levels of tTG and Hsp90 in whole cell lysates of the PDX samples HCI-001-HCI-003, and HCI-005-HCI-010). We then screened the conditioned medium from the different breast cancer PDXs to identify those samples capable of activating VEGFRs in a Bevacizumab-insensitive manner. The stimulatory capability of the conditioned medium from a number of the samples was insensitive to Bevacizumab (that is, samples HCI-001, −002, −005, −007,

−010 and −013) (Fig. 7a), with the lack of sensitivity to the drug being correlated with the detection of secreted $VEGF_{90K}$ (Table 2).

PDX samples HCI-001 and HCI-002, which were Bevacizumab-insensitive but benefitted from the combination of an Hsp90 inhibitor with the VEGF drug when assaying tumour growth (Fig. 1a,b) and blood vessel formation (Supplementary Fig. 1A,B), yielded MVs with associated $VEGF_{90K}$ (Fig. 7b). As anticipated, the ability of their MVs to stimulate VEGFR activation was unaffected by Bevacizumab (Fig. 7c), whereas when MVs were treated with 17AAG, sensitivity to the VEGF antibody was restored (Fig. 7c). On the other hand, tumour growth by the PDX sample HCI-003, that secreted low amounts of $VEGF_{90K}$ (Supplementary Fig. 7A; Fig. 7a), was inhibited by Bevacizumab alone, with no additional benefits occurring from the combination of Bevacizumab and 17AAG (Fig. 7d).

**Discussion**
The roles of EVs in a variety of biological contexts and disease states are attracting significant interest because of the unique mechanisms of cell-cell communication that they offer, and for their potential use as biomarkers and diagnostic reagents. Both MVs and exosomes have been suggested to exert major functions in a number of stages of tumorigenesis[5–7,9–15,42]. These include crafting the architecture for the tumour microenvironment, and the education of bone marrow-derived cells to help in the creation of the pre-metastatic niche. EVs have also been implicated in tumour angiogenesis, with an earlier report showing that MVs derived from certain cancer cells containing activated EGF receptors caused the up-regulation of VEGF expression in endothelial cells and an accompanying activation of VEGFRs[12]. Here, we report a mechanism by which MVs shed from breast cancer cells trigger an immediate and sustained activation of VEGFRs on endothelial cells that is insensitive to the inhibitory actions of VEGF antibodies like Bevacizumab.

The ability of breast cancer cell-derived MVs to stimulate VEGFR-signalling is dependent on VEGF that is associated with the vesicles, as knockdowns of VEGF in cancer cells from which the MVs originate eliminate their stimulatory activity. In addition to the classical forms of VEGF, breast cancer cells also generate and secrete a larger VEGF species with an apparent size of 90 kDa ($VEGF_{90K}$). It is $VEGF_{90K}$ that we routinely detect in MVs. $VEGF_{90K}$ is not a unique splice variant form of VEGF but is generated through the crosslinking of $VEGF_{165}$ by tTG (Fig. 8a). This most likely occurs once these proteins are accessible to the outside of the cell, that is,

**Figure 6 | VEGF released from MVs regains its sensitivity to Bevacizumab.** (**a**) Non-permeabilized MDAMB231 cells were analysed by immunofluorescent confocal microscopy using anti-VEGF and anti-Hsp90 antibodies. Top images: $VEGF_{90K}$ and Hsp90 are detected on MVs (arrows). Bottom images: MDAMB231 cells treated with 10 μM 17AAG overnight were fixed and stained. Scale bar, 10 μm. Far-right: Blow-ups of the MVs. (**b**) MDAMB231 cell MVs treated without (lane 2) or with (lane 3) 17AAG at 37 °C for 2 h were lysed and immunoprecipitations were performed using a Hsp90 antibody (25 μg MV protein, each). (**c**) $VEGF_{90K}$ was generated by tTG-catalysed crosslinking of $rVEGF_{165}$, incubated with Hsp90 (30 ng), either without (lane 1) or with 17AAG (10 μM) (lane 2), at 37 °C for 1 h and immunoprecipitated using an anti-Hsp90 antibody. (**d**) MDAMB231 cell MVs treated without (lane 1) or with 17AAG (lane 2) at 37 °C for 2 h were collected on a 0.22 μm filter. The filtered MVs and the flow-through were immunoblotted. (**e**) Serum-deprived HUVECs were untreated (lane 1), or exposed to $VEGF_{90K}$ ($\sim$10 ng ml$^{-1}$) released from 17AAG-treated MVs and present in the flow-through from the experiment shown in Fig. 5d, in the presence (lane 2) or absence (lane 3) of Bevacizumab, for 15 min, lysed and immunoblotted. (**f**) Serum-starved HUVECs, untreated (lanes 1, 2, 5 and 6) or pre-treated with 10 μM GA for 1 hour (lanes 3 and 4), were incubated without (lane 1) or with MVs (5 μg ml$^{-1}$ protein) from MDAMB231 cells (lane 2), together with 200 ng ml$^{-1}$ pan inactivating VEGF antibody (lane 4), 10 μM 17AAG (lane 5) or both the VEGF antibody and 17AAG (lane 6), for 15 min. Cell extracts were lysed and immunoblotted. (**g**) Serum-starved HUVECs were incubated with the $VEGF_{90K}$–Hsp90 complex ($\sim$10 ng ml$^{-1}$) without (lane 1) or with Bevacizumab (0.5 μg ml$^{-1}$) (lane 2) for 15 min, lysed and immunoblotted. (**h**) Left: Relative amounts of tubulogenesis for HUVECs untreated (control; histogram 1) or treated with the $VEGF_{90K}$–Hsp90 complex ($\sim$20 ng ml$^{-1}$ total protein) without (histogram 3) or with 0.5 μg ml$^{-1}$ Bevacizumab (histogram 4). Right: Images of the tubulogenesis assays.

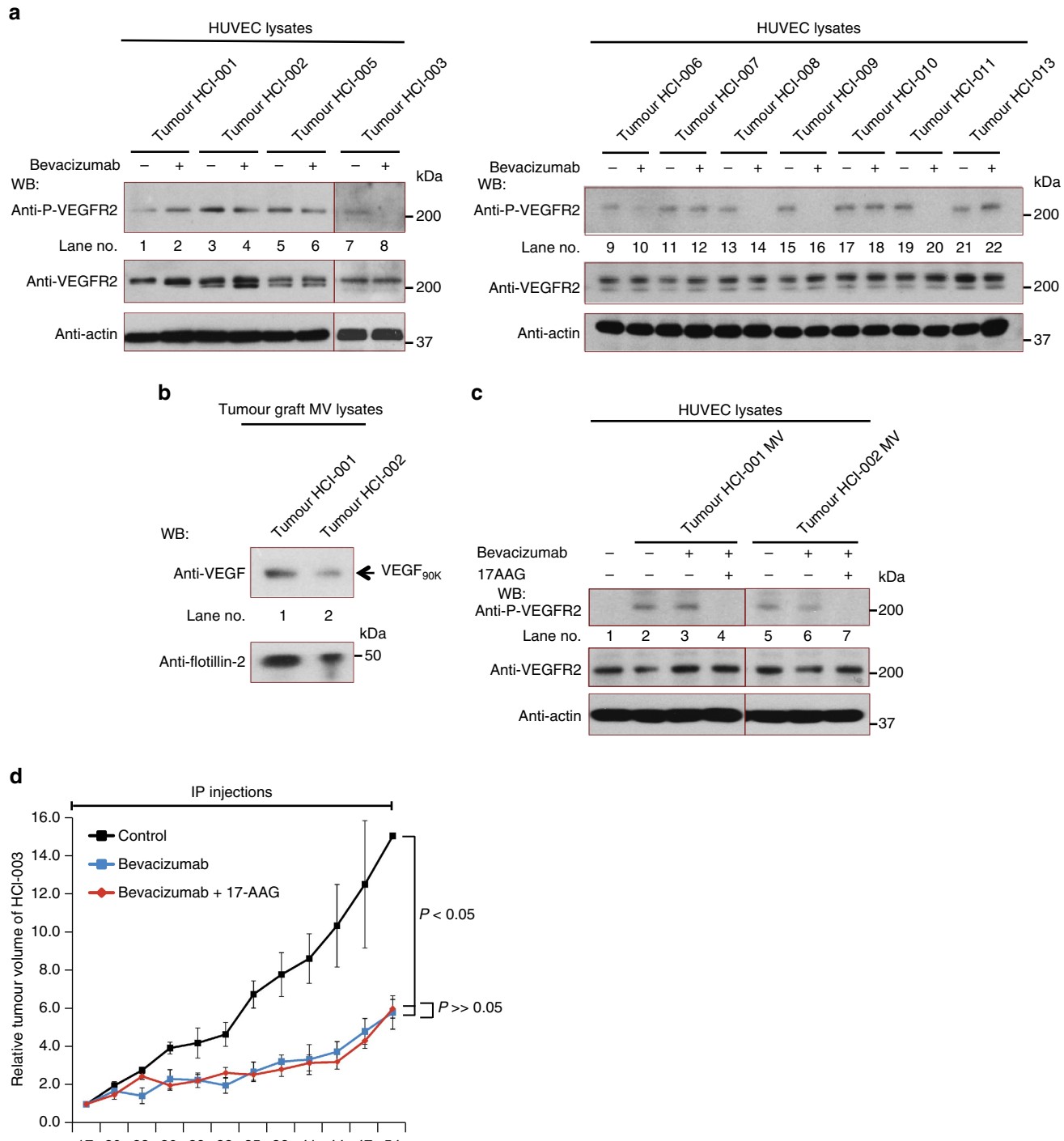

**Figure 7 | The role of MVs in Bevacizumab insensitivity extends to PDXs.** (**a**) Lysates of serum-deprived HUVECs treated with conditioned medium (CM; 20 µg ml$^{-1}$ total protein) from cell cultures of PDX samples HCI-001-003 and HCI-005 (Left) or HCI-006-011 and HCI-013 (Right), described in Table 1, were supplemented without or with 0.5 µg ml$^{-1}$ Bevacizumab, as indicated, for 15 min. HUVEC lysates were immunoblotted with antibodies that recognize phosphorylated VEGFR2, total VEGFR2 or actin. (**b**) MVs from cultures of cells established from PDX samples HCI-001 and HCI-002 were isolated, lysed and immunoblotted with antibodies against pan VEGF or the MV marker flotillin-2. (**c**) Serum-deprived HUVECs were incubated with serum-free medium alone (control; lane 1), or with serum-free medium containing the indicated combinations of MVs from PDX samples HCI-001 or HCI-002 (5 µg ml$^{-1}$ of MV protein), 0.5 µg ml$^{-1}$ Bevacizumab (lanes 3 and 6) and 10 µM 17AAG (lanes 4 and 7), for 15 min and then lysed. Cell extracts were immunoblotted with antibodies that recognize phosphorylated VEGFR2, total VEGFR2 or actin. (**d**) Plots showing relative mean tumour volumes (mm$^3$) of tumour graft HCI-003 in NOD/SCID mice that were untreated (vehicle control only), or treated with either Bevacizumab or Bevacizumab plus 17AAG. The differences between the tumour volumes for the Bevacizumab treatment group versus the vehicle only (control) group were statistically significant ($P < 0.05$).

**Table 2 | Correlation between secreted VEGF$_{90K}$ levels and Bevacizumab resistance in tumour grafts.**

| Breast tumour sample ID | Secreted VEGF$_{90K}$ level | Bevacizumab resistance |
|---|---|---|
| HCI-001 | High | Yes |
| HCI-002 | High | Yes |
| HCI-003 | Low | No |
| HCI-005 | High | Yes |
| HCI-006 | Low | Partly |
| HCI-007 | Medium | Yes |
| HCI-008 | Low | No |
| HCI-009 | Low | No |
| HCI-010 | Medium | Yes |
| HCI-011 | Low | No |
| HCI-012 | Low | ND |
| HCI-013 | High | Yes |

ND, not determined.
Table summarizing our findings regarding the correlation between the levels of VEGF$_{90K}$ secreted from the PDX samples and Bevacizumab resistance. Conditioned medium was prepared from cells cultured from the indicated tumour graft samples. Each sample was normalized for total tumour graft protein and assayed for VEGFR activation in HUVECs by VEGFR2 auto-phosphorylation, in the presence and absence of Bevacizumab (0.5 µg ml$^{-1}$).

the location where tTG is most active as a crosslinking enzyme[7]. We have examined whether MV-associated VEGF$_{90K}$ secretion is increased under hypoxic conditions by culturing MDAMB231 and HCI-002 cells in $CoCl_2$ (ref. 51). While these conditions enhanced total VEGF secretion from the cells (see the right-hand plots in Supplementary Fig. 8A,B), they neither increased the biogenesis of MVs nor the secretion of MV-associated VEGF$_{90K}$ (see the left-hand panels in Supplementary Fig. 8A,B). We have also examined the levels of VEGF$_{90K}$ in cell lysates, on treating the cells with 17AAG, and observed that after 30 min of treatment, there was an ~2-fold accumulation of VEGF$_{90K}$, suggesting that by blocking the interaction of VEGF$_{90K}$ with Hsp90, the amount of VEGF$_{90K}$ secreted via MVs was reduced (Supplementary Fig. 8C). Interestingly, this accumulation of VEGF$_{90K}$ was not sustained, implying that under conditions where cells undergo a persistent treatment with the Hsp90 inhibitor, VEGF$_{90K}$ may ultimately be secreted through an alternative mechanism which is not dependent on its interaction with Hsp90.

The association of VEGF$_{90K}$ with MVs makes it markedly less susceptible to the inhibitory actions of either Bevacizumab or a pan VEGF antibody. Bevacizumab appears to bind much more weakly to MV-associated VEGF$_{90K}$, compared with VEGF$_{90K}$ that is not associated with MVs. Apparently, the epitope sites on MV-associated VEGF$_{90K}$ are not completely inaccessible, as vesicle-bound VEGF$_{90K}$ can be visualized using VEGF antibodies by immunofluorescence, although this requires antibody concentrations at least 50-fold greater than those typically used in functional assays or in clinical applications. The combination of a weak affinity for MV-associated VEGF$_{90K}$, and the potential for multivalent interactions between the oligomeric VEGF$_{90K}$ molecules on the vesicle surface and the VEGFRs on endothelial cells, most likely accounts for Bevacizumab being ineffective in blocking the stimulatory activity of the MVs. The interaction of MV-associated VEGF with VEGFRs might make it difficult for these receptors to undergo endocytosis and/or degradation, at least on a normal time scale, which could explain the sustained signalling by VEGFRs stimulated by MVs, compared with the transient signals triggered by VEGF in the absence of MVs. However, other potential mechanisms for the sustained signalling by MVs are also possible. Studies by Al-Nedawi et al.[12] suggested that MVs shed by cancer cells expressing high levels of EGF receptors stimulated the expression of VEGF, giving rise to sustained signalling by endogenous VEGF from an endomembrane site (for example, endosomes), which

could have important consequences given the suggestions that the endocytosis and localization of VEGFR2 to endosomes can play a key role in signalling to its downstream partners[52–54].

The association of VEGF$_{90K}$ with MVs is mediated through its interaction with at least one other cargo protein, Hsp90 (as depicted in Fig. 8a,b). Thus, treating MVs with Hsp90 inhibitors such as 17AAG causes the release of VEGF$_{90K}$ from the vesicles, restoring its sensitivity to Bevacizumab (depicted in Fig. 8b, bottom half of the scheme). This could explain how the combination of 17AAG and Bevacizumab provided a significant benefit when assaying both tumour growth and blood vessel formation in mouse xenografts, where the cancer cells being examined (that is, MDAMB231 cells and cells cultured from PDXs) shed relatively large amounts of MVs containing VEGF$_{90K}$. However, blocking Hsp90 function can impact a number of aspects of cancer cell survival, which may contribute to the inhibition of tumour growth, particularly when combined with a drug (Bevacizumab) that ultimately leads to nutrient deprivation and cellular stress. It remains to be seen whether such a combination might ultimately provide therapeutic benefits, particularly given that Bevacizumab has been used in a number of cancer clinical trials and treatment regimens[18,20–25], but has often failed, as have other anti-angiogenesis strategies, for a variety of reasons including the development of resistance. Indeed, there appear to be multiple contributors to the limitations observed with Bevacizumab and other angiogenesis drugs[24,52,55,56]. Our findings that MVs shed by breast cancer cells are insensitive to Bevacizumab at the levels used in functional assays and in therapeutic applications, offer an additional mechanism that might account for some of the ineffectiveness of this VEGF antibody in cancer treatment. Moreover, they highlight the importance, when designing drug treatment regimens, of taking into account not only the traditional soluble factors secreted by aggressive cancer cells, but also the fact that tumour cells shed MVs which can significantly compromise the success of such strategies.

There is still a good deal to learn regarding the detailed mechanisms by which MV-associated VEGF$_{90K}$ activates VEGFRs and stimulates the necessary signals that contribute to tumour angiogenesis. This includes determining how Hsp90 tightly associates with the surface of MVs, as well as how it binds to VEGF$_{90K}$. It will be important to understand why some cancer cells generate and shed relatively large amounts of MVs with associated VEGF$_{90K}$, while others are much less effective in producing these vesicles. Such differences might

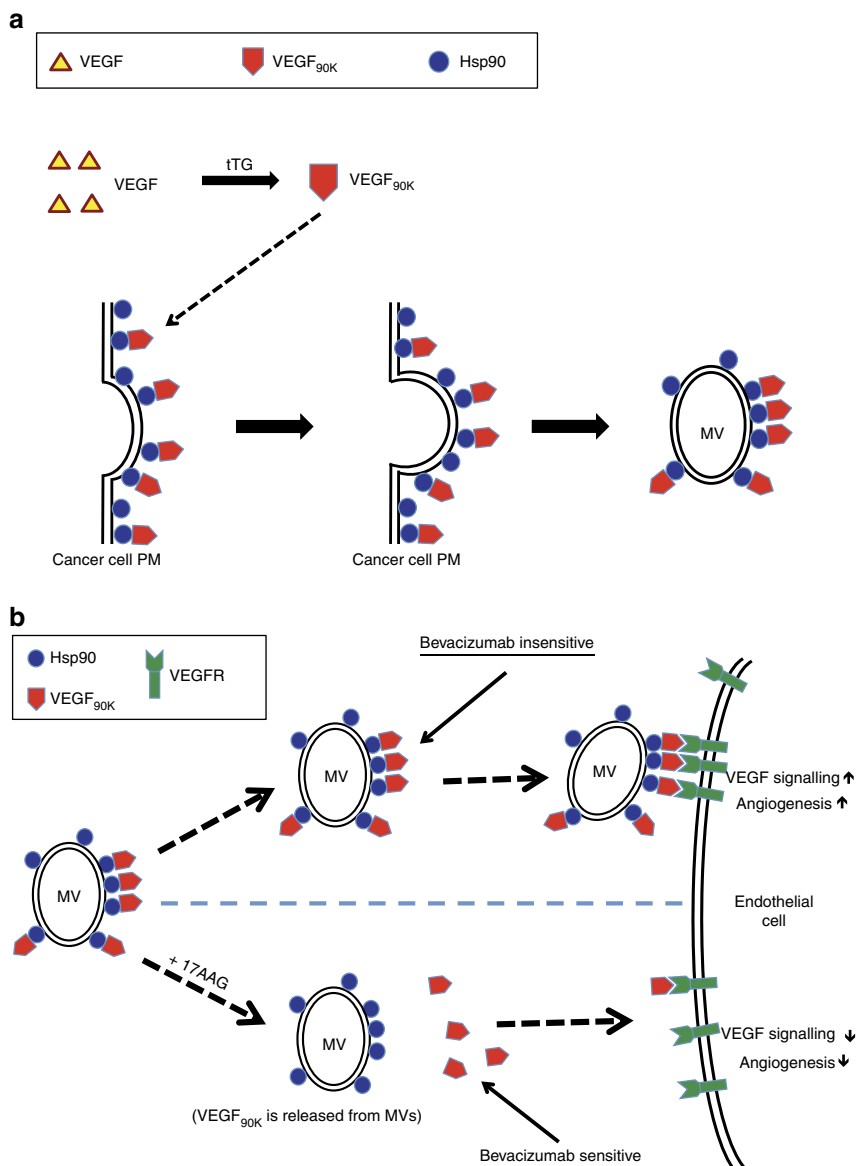

**Figure 8 | Diagrams depicting cancer cells generating VEGF_{90K} and shedding MVs with associated VEGF_{90K}.** (**a**) VEGF$_{165}$ is crosslinked by tTG to generate VEGF$_{90K}$ (top). MVs with associated VEGF$_{90K}$ are budding and shed from cancer cell plasma membrane (bottom). (**b**) Diagram depicting cancer cells shedding MVs with associated VEGF$_{90K}$ that engage recipient endothelial cells and activate VEGFRs, thereby promoting angiogenesis. MV-associated Hsp90 binds to VEGF$_{90K}$, enabling the vesicles to activate VEGFRs on endothelial cells and stimulate the formation of new blood vessels. This stimulation is insensitive to Bevacizumab (top). 17AAG causes the release of VEGF$_{90K}$ from MVs. The free VEGF$_{90K}$ activates VEGFRs and stimulates angiogenesis but is sensitive to the inhibitory actions of Bevacizumab (bottom).

reflect the extent to which RhoA is activated, given that RhoA-dependent actin filament organization is required for the maturation of VEGF$_{90K}$-associated MVs along the surfaces of cancer cells[42]. The ability of cancer cells to generate MVs containing VEGF$_{90K}$ might also reflect the relative expression and activation of tTG, as well as the relative expression levels of Hsp90. It is interesting that exosomes are devoid of VEGF$_{90K}$, although one plausible possibility for these findings is that VEGF$_{165}$ is delivered to the outer surfaces of cells via a classical secretory pathway, where it then encounters tTG on the maturing MVs budding from these membrane sites. This would be consistent with our finding that brefeldin A, which blocks the classical intracellular trafficking pathway, inhibits the formation of VEGF$_{90K}$ (Q. Feng, unpublished results). In the future, it will also be important to determine whether MVs with

associated VEGF$_{90K}$ can be isolated from the blood of human patients with breast cancer, as this could ultimately provide diagnostic benefits. Finally, we will want to examine the benefits of this type of combination therapy in models of metastatic breast cancer[57,58], where anti-angiogenic therapies have often been poorly effective. While the answers to these and other questions surrounding MV function await future investigations, it is now evident that the actions of MVs, as well as exosomes, in cancer cell biology and tumour progression represent exciting new areas that could significantly impact future basic and translational research efforts.

## Methods
**Isolation of MVs.** Conditioned medium from $\sim 2 \times 10^7$ serum-starved MDAMB231, SKBR3 or HeLa cells, as well as U87 and HT29 cells, was collected

and partially clarified by 3 consecutive centrifugations at 1,000 r.p.m. ($\sim 300g$) for 5 min, or by filtration using a Millipore ultrafree PVDF filter with a 3.1 μm pore size. To generate intact MVs for functional analyses, partially clarified conditioned medium was filtered using a Millipore ultrafree PVDF filter with a 0.1 μm or 0.22 μm pore size. MVs retained by the PVDF membrane were rinsed with PBS and resuspended in serum-free medium.

MVs were isolated from tumour grafts as follows. Portions of tumour grafts were trypsinized, mechanically disrupted, and placed in 100 mm dishes for 3 h in RPMI medium containing 10% FBS. The cells ($\sim 2 \times 10^7$) that attached were then starved in serum-free RPMI medium for 4 h, at which point the MVs from the conditioned media were isolated.

**Animal tumour models.** All experiments involving mice were carried out according to the protocols approved by the Animal Care and Use Committees at Cornell University and the University of Utah. For the xenograft experiments, MDAMB231 breast cancer cells ($3 \times 10^6$) were injected subcutaneously into the two flanks (4 mice) of each 4–6-week-old female NIH-III nude mouse (Charles River, Wilmington, MA, USA). When tumours of 1–2 mm in diameter were detected, the mice were randomly divided into four groups (4 mice for each group, each doubly injected) and IP injections with drugs (5 mg Bevacizumab per kg animal weight; 20 mg 17AAG per kg animal weight) or the vehicle control (RPMI containing 4% DMSO, v/v), were initiated and performed every third day for 15 days. Tumour sizes were calculated using the formula: [length (mm) × width (mm) × width (mm) × 0.5]. On day 24, the resulting tumours for the different treatment groups were excised and weighed.

For human breast cancer tumour graft experiments, tumour grafts derived from breast cancer patients[27] were cut into pieces (2 mm × 2 mm × 2 mm) and implanted into the mammary glands of NOD/SCID mice. When the tumour grafts reached $\sim 3$ mm in diameter (2 weeks after implantation for HCI-002 and 4 weeks after implantation for HCI-001), the mice were randomly divided into four groups and IP injections with drugs (2.5 mg Bevacizumab per kg animal weight; 10 mg 17AAG per kg animal weight) or the vehicle control (RPMI containing 4% DMSO + 5% Cremophor EL (Sigma, St Louis, MO, USA) + 5% ethanol, v/v), were initiated and performed every other day for 4 weeks. Tumour sizes were calculated as described above. After 4 weeks of treatment, tumours from mice in each group were excised and weighed. The tumour graft samples were then fixed, sectioned and stained with a PECAM-1/CD31 antibody to detect endothelial cells/blood vessels by the Immunopathology Laboratory at Cornell University.

**In vivo angiogenesis assays.** The directed in vivo Angiogenesis Assay Kit (Trevigen, Gaithersburg, MD, USA) was used to assay endothelial cell recruitment in mice according to the manufacturer's protocol. Briefly, implant grade silicone cylinders that are closed at one end (angioreactors) were filled with 20 μl of BME gel premixed with 2 μl of RPMI (vehicle control), or with or without MDAMB231 cells, rVEGF$_{165}$ and/or inhibitors. The angioreactors were then implanted subcutaneously in the dorsal flanks of nude mice. One week later, the angioreactors were removed. The vascular endothelial cells that migrated (invaded) into the angioreactors were quantified using the FITC-Lectin Detection protocol[28].

**Tubulogenesis assays.** The in vitro Angiogenesis Assay Tube Formation Kit (Trevigen) was used according to the manufacturer's instructions. Briefly, BME solution was added into 96-well plates (50 μl per well), and the plates were incubated at 37 °C for 60 min. HUVECs were diluted with Endothelial Basal Medium in the presence or absence of rVEGF$_{165}$, MVs and/or inhibitors. HUVECs ($1 \times 10^4$ cells) were then added to each well containing the gelled BME. The plates were incubated at 37 °C in a CO$_2$ incubator for 4 h, at which time the HUVEC-formed tubular networks were visualized using a light microscope.

**Data analyses.** For tubulogenesis assays: Photographs of the cell cultures were taken and the lengths of the tubes that formed for each condition were determined using ImageJ (NIH). The data shown represents the mean ± s.d. from at least three independent experiments. For in vivo angiogenesis assays: The data was expressed in relative fluorescent units (RFUs). Relative invasion = test sample (RFU)/negative control (RFU). The data shown represents the mean ± s.d. from 4 to 8 angioreactors. For animal tumour models: The amounts of endothelial cells for each condition were determined using ImageJ software and the relative endothelial cell counts and blood vessel numbers for samples were plotted. The data shown represents the mean ± s.d. from 6–10 sample sections (8–12 views were counted for each section). All P-values were determined by the student's t-test and $P < 0.05$ is considered as significant difference.

**Reagents.** The pan VEGF antibody that recognizes all forms of VEGF-A was obtained from Santa Cruz (Dallas, TX, USA) (SC-507; used at 1:2,000). The actin (MABT825; used at 1:5,000), IκBα (07-1483; used at 1:2,000), CD-63 (CBL553; used at 1:2,000) and the VEGF$_{165}$ (07-1419; used at 1:2,000) antibodies, were from Millipore (Billerica, MA, USA). The HIF-1α antibody (NB100-105; used at 1:2,500) was from Novus Biologicals (Littleton, CO, USA), while the flotillin-2 (3436; used at 1:2,000), Hsp90 (4877; used at 1;1,000), VEGFR2 (9698; used at 1:2,500),

phospho-VEGFR2 (3770; used at 1:2,500), phospho-ERK (4370; used at 1:2500), and ERK (9102; used at 1:2,500) antibodies were from Cell Signaling (Danvers, MA, USA). MDC, 17AAG and the tTG (3557; used at 1:1,000) antibody were from Invitrogen (Waltham, MA, USA). Geldamycin was from Cell Signaling and Bevacizumab was a kind gift from Dr Xu Peng (Texas A&M, College Station, TX, USA). Recombinant VEGF$_{165}$ was from Thermo (Waltham, MA, USA), recombinant Hsp90 was from Prospec, East Brunswick, NJ, USA and recombinant tTG was prepared as described previously[59]. The Steriflip PVDF filters (0.1 or 0.22 μm pore size) and centrifugal filters, were from Millipore.

**Immunohistochemistry.** Sections (4 μm thick) of formalin-fixed/paraffin-embedded human tumours excised from mice were used for immunohistochemical analysis. After de-paraffinization in xylene and rehydration in graded ethanol, antigen retrieval was performed by heating the sections in citrate buffer (10 mM, pH 6.0) for 5 min. Non-specific staining was blocked with a mixture of 10% goat serum and casein (2 ×) for 30 min at room temperature. The slides were incubated in rabbit polyclonal anti-PECAM-1 (M-20) antibody (Santa Cruz) (1:100 diluted in PBS containing 1 × casein) for 1.5 h at room temperature, followed by Texas Red conjugated goat anti-rabbit IgG antibody (Invitrogen) at 1:200 in PBS for 30 min at room temperature. Rabbit IgG was used as a negative control. Finally, the slides were mounted in Vectashield mounting media containing DAPI (Vector Laboratories, Burlingame, CA, USA).

**Immunofluorescence.** Cells were fixed with 3.7% paraformaldehyde and then permeabilized with PBS containing 0.1% Triton X-100. Samples were incubated with the primary antibodies and then with Rhodamine- or Oregon green 488-conjugated secondary antibodies. Rhodamine-conjugated phalloidin was used to stain (F)-actin filaments, and DAPI was used to label nuclei. Fluorescent microscopy images were captured and processed using IPLab software (Scientific Instrument Company, Campbell, CA, USA).

**Confocal microscopy.** Cells were fixed in 4% paraformaldehyde in PBS for 15 min and stained for the indicated proteins. Images were obtained with a Zeiss LSM 510 Meta confocal microscope (Carl Zeiss, Germany) using a Plan-Apochromat × 63/1.40 oil objective. The pinhole size was set to 1 airy unit for all channels. Z-stack images were collected at optimal intervals as suggested by the ZEN software (Zeiss), and the three-dimensional images were produced using Volocity software (Perkin-Elmer, Waltham, MA, USA).

**RT-PCR.** Total RNA was isolated and reverse transcription was carried out using the SuperScript III First-Strand Synthesis System (Invitrogen). The VEGF sequence containing a portion of the 5′ and 3′ UTR sequences was amplified using the VEGF-forward primer: GGGGACAAGTTTGTACAAAAAAGCAGGCTTCCGGCGTGAGCCCTC CCCCTTG and the VEGF-reverse primer: GGGGACCACTTTGTACAAGAAAG CTGGGTGGATCTGGTTCCCGAAACCCTG.

VEGF transcript variants were detected by using a common forward primer (Exon3F: CCCTGATGAGATCGAGTACATCTT) and reverse primers covering the exon 5 and 7 junction (Exon5/7R: AGCAAGGCCCACAGGGATTT), the exon 5 and 8 junction (Exon5/8R: GCCTCGGCTTGTCACATTTT), exon 6 (Exon6R: AACGCTCCAGG ACTTATACCG) or exon 8 (Exon8R: ACCGCCTCG GCTTGTCAC).

**tTG-catalysed crosslinking of rVEGF$_{165}$ to generate VEGF$_{90K}$.** rVEGF$_{165}$ (100 ng) was incubated with recombinant tTG (100 ng) in a buffer containing 40 mM CaCl$_2$ and 40 mM dithiothreitol for up to 1 hour on ice. The crosslinked rVEGF$_{165}$ (that is, recombinant VEGF$_{90K}$) was then used in immunoprecipitation experiments or was added to assays monitoring VEGFR activation and signalling.

**Data availability.** The authors declare that the data supporting the findings described in this study are available within the article and its Supplementary Information files (Uncropped scans of western blots were included in Supplementary Figure 9), or available from the authors upon request.

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

## Acknowledgements

We acknowledge members of the Cerione laboratory as well as Drs. Scott Emr and David Holowka, for critically reading the manuscript, Cindy Westmiller for excellent technical assistance, and Dr Elizabeth Buckles and Lynn Dong in the Immunopathology Laboratory, Cornell University, for performing IHC and analysing possible metastatic nodules.

## Author contributions

Q.F. performed the majority of the experiments. C.Z. performed the RT-PCR, MV isolation and confocal microscopy experiments. J.E.D., B.B., D.L., C.Z. and Q.F. performed the mouse experiments. Q.F., M.A., K.F.W., A.W. and R.A.C. Analysed the data and wrote the manuscript.

## Additional information

**Competing financial interests**: The authors declare no competing financial interests.

**How to cite this article**: Feng, Q. *et al.* A class of extracellular vesicles from breast cancer cells activates VEGF receptors and tumour angiogenesis. *Nat. Commun.* **8**, 14450 doi: 10.1038/ncomms14450 (2017).

