## [Peer Review File · Nature Communications]

Reviewers' comments:

Reviewer #1 (Remarks to the Author):

This is a well written and well executed study describing a novel mechanism of pro-angiogenic signalling which may, potentially, help to explain resistance to the anti-angiogenic drug bevacizumab in breast cancer. I think that the work is technically sound and the experimental data would appear to support main conclusions of the study.

As the authors point out, bevacizumab has failed to make an appreciable impact in the treatment of breast cancer. It is important that we understand why, because this may lead to new treatment options for targeting the tumour vasculature in breast cancer. Therefore, this manuscript addresses an important question.

Specifically, the authors show that breast cancer cells can secrete a 90 kDa species of VEGF that is formed due to the crosslinking activity of transglutaminase. Moreover, they show that this 90 kDa species of VEGF is coupled to extracellular vesicles (EV) by HSP90. Interestingly, whilst these EVs show potent pro-angiogenic activity both in vivo and in vitro, this pro-angiogenic activity is not blocked by bevacizumab. The authors propose that this may be one mechanism through which bevacizumab activity is limited in breast cancer. Moreover, they show that this resistance mechanism is overcome by combining bevacizumab with HSP90 inhibitors, because the HSP90 inhibitors trigger the release of 90 kDa VEGF from EVs - thus allowing bevacizumab free access to block VEGF. The authors therefore propose that VEGF and HSP90 inhibitors should be combined in patients.

I have the following comments:

(1) Page 7, line 141. When the authors say "the VEGF drug" I think they should replace this with the word "bevacizumab" or "anti-VEGF antibody" because the term "the VEGF drug" is imprecise. Same issue should be corrected on page 15 line 327.

(2) Page 10, line 219. The authors say "Bevacizumab and other anti-VEGF antibodies" when referring to Fig 4D. In this figure, they only test ONE other anti-VEGF antibody, but the sentence infers that more than one other anti-VEGF antibody was tested. I believe that this is misleading and needs to be corrected. Same issue should be corrected on page 15 line 338.

(3) Page 15, line 326. The authors say that their mechanism "does not involve nor require the upregulation of VEGF in the targeted endothelial cells." However, I could not find the experimental evidence where the authors prove this? Could the authors make it clear what evidence they provide to support this statement? If there is no evidence presented in the paper, it may be better to remove this sentence from the manuscript.

(4) Page 16, line 348-349. I have a problem with the author's hypothesis that the trapping of VEGFR2 at the cell surface may lead to an increased pro-angiogenic signalling of VEGFR2. There are several studies showing that, in fact, the endocytosis of VEGFR2 is required for signal transduction to downstream pathways and the full activation of angiogenesis:

Lanahan et al

VEGF receptor 2 endocytic trafficking regulates arterial morphogenesis.

PMID: 20434959

Sawamiphak et al

Ephrin-B2 regulates VEGFR2 function in developmental and tumour angiogenesis.
PMID: 20445540

Gourlaouen et al
Essential role for endocytosis in the growth factor-stimulated activation of ERK1/2 in endothelial cells.
PMID: 23341459

Whilst it is fine for the authors to speculate that trapping at the cell surface protects the receptor from degradation / dephosphorylation, they should add a brief note to the Discussion acknowledging that endocytosis of VEGFR2 may in fact be required to deliver the full pro-angiogenic signal to endothelial cells (and these three papers above should be cited as evidence of this fact).

(5) Related to the above. Is it possible that the EVs are endocytosed more effectively by the endothelial cells (compared to free VEGF) or signal more effectively from endosomes in endothelial cells and that this explains the heightened signalling by EV-coupled 90 kDa VEGF? I do not propose that the authors perform further experiments to address this, but I do propose that some text regarding the potential for EVs to promote VEGFR2 signalling from endosomes should be added. However, the authors are free to ignore this comment if EVs are, in fact, too large to be effectively endocytosed.

(6) Page 16, lines 360-361. The authors state correctly that there are other mechanisms of resistance to anti-VEGF therapy. Although reference 24 is a good one, it is now rather old and there are more recent reviews of this rapidly moving field. In addition to citing reference 24, I suggest to also cite these additional more recent references that discuss mechanisms of resistance to anti-VEGF therapy (see below). I also suggest to remove reference 51, as this paper does not actually address the mechanisms of resistance to anti-VEGF drugs in breast cancer.

Ebos et al (2011)
Antiangiogenic therapy: impact on invasion, disease progression, and metastasis.
PMID:21364524

Vasudev et al (2014)
Anti-angiogenic therapy for cancer: current progress, unresolved questions and future directions.
PMID: 24482243

(7) Page 16 - 17. It is good to see that the authors discuss the limitations of their study and the questions that still remain unanswered, because these comments highlight important areas for further research. However, I propose that they need to include two further points within the Discussion section of the article:

Firstly, unless I am mistaken, they have not proven the presence of VEGF-90K vesicles in human tumours. I understand that they have shown their existence in cultured patient-derived cancer cells, but can these same vesicles be isolated from fresh human tumour specimens of breast cancer (without the culture step) or from the blood of humans with breast cancer? Authors should state that they did not demonstrate this in the current study and that this is something that should be examined in the future.

Secondly, the authors experiments only address the effect of the bevacizumab + HSP90 combination on subcutaneously implanted tumours (i.e. a model that roughly equates to primary breast cancer). However, several studies have now shown that, whilst anti-angiogenic therapies can be effective against models of primary breast cancer, the same anti-angiogenic therapies are very poorly effective

(or sometimes completely ineffective) in models of metastatic breast cancer (e.g. Ebos et al Cancer Cell PMID: 19249681, e.g. Guerin et al Cancer Research PMID: 23610448) Authors should state that they did not demonstrate efficacy of bevacizumab + HSP90 combination in metastatic breast cancer models in the current study and that this is something that should be examined in the future.

Reviewer #2 (Remarks to the Author):

The manuscript by Feng et al. deals with the interesting topic of Bevacizumab resistance in breast cancer. By using the triple negative breast cancer cell line MDAMB231, the authors show that VEGF165 cross links with help of tTG enzyme and creates VEGF90K which is subsequently packed into microvesicles with help of Hsp90 and is shed out of cells and cause resistance to Bevacizumab. The authors have validated their hypothesis using another breast cancer cell line SKBR3 and cervix adenocarcinoma HeLa cells as negative control. Results are further validated in responsive and refractory patient derived materials. Authors confirm the in vitro findings with in vivo tumor growth curves using tumor cell lines and also patient derived material however some more complementary experiments are required to shed light to this new concept of drug resistance.

1) Reviewer was wondering if the MVs containing VEGF90K are only specific to breast cancer or if they can be also found in other cancer types? Could the author present some Western blots from other types of cancers that are also resistant to Bevacizumab in order to clarify the specificity of these vesicles to breast cancer.

2) Reviewer would like the author to be more thorough about the nature of the cross-linked VEGF90K. Why and how does the cross-linking of VEGF create the 90K VEGF and not any other molecular weight. Do the authors suggest that there is always a very defined number of crosslinked VEGF molecules? Furthermore, can the authors rule out that the found protein of 90 kDa is a result of crosslinking VEGF-A and Hsp90 or other VEGF-A interacting proteins?

3) Due to the correlation between the intra-tumoral hypoxia and VEGF levels in tumor microenvironment, reviewer suggests to show hypoxic regions under different conditions to be displayed in supplementary figure S1 A. Furthermore, the authors should analyze basic features of treated and control tumors, such as hypoxic regions, blood vessel density, blood vessel perfusion, and metastatic index.

4) Due to the vast physiological functions of Hsp90, how can the author exclude the function of 17AAG treatment merely on the Hsp90s binding to the VEGF90K and not all the endogenous Hsp90s present within the cell? How can the authors show that the inhibition with 17AAG, does not affect and interfere with the general physiological functions of the Hsp90 protein?

5) Regarding figure 2G, reviewer suggests adding another control group containing only rVEGF165+17AAG in order to investigate if there would still be a formation and accumulation of VEGF90K in the presence of tTG without Hsp90? And as a follow up experiment, in figure 2I, the reviewer suggests to add a control group containing HUVEC cells cultured together with rVEGF165 +17AAG.

6) Reviewer believes that the image in figure 3C could be improved by using a more specific cell surface marker or using a cell membrane marker (dye) to visualize the cell surface together with Actin specific marker with better visual quality to show the MVs containing VEGF90K.

7) In order to clarifying the specific role of tTG in cross linking and generating the VEGF90KD, it is

necessary to include a Western Blot band in figure 3H to show the lack of VEGF90K upon knock down of the tTG or in presence (control) of the enzyme.

8) In order to show the specific function of tTG in cross-linking the VEGF165 and create VEGF90K prior to its binding to Hsp90, reviewer recommends to inhibit Hsp90 and show that no VEGF90K will be found in the shedding MVs. Also, to show the predicted accumulation of VEGF90K in the cytoplasm upon inhibition of Hsp90, it is suggested to perform a Western Blot on the cell lysate from the cells treated with 17AAG.

9) Due to increase in VEGF165 levels under hypoxic conditions, do the authors expect increased levels of VEGF90K also in these conditions?

In figure 4B, when on the right panel, MDAMB231 MVs, why the levels of VEGF90K does not go down after 45 min? Has the author checked longer time points? Is there any accumulation of the VEGF90K which leads to stabilizing the protein after 45 min?

10) Reviewer suggests that the authors add two negative controls for figure 5A and 6A, i) VEGF KD; ii) Incubate the sample with IgG control and secondary antibody to eliminate the possibility of an unspecific secondary antibody signal.

11) Regarding figure 5C, in order to show the crosslinking effect of tTG on VEGF165, reviewer wish to suggest an additional control as following:
rVEGF165 + rtTG + rHsp90 - (basically rVEGF165 + rtTG in absence of rHsp90).

12) Reviewer wonders why in figure 6G, there is no stimulation of VEGFR2 when HUVEC cell lysate is incubated with VEGF90K-Hsp90 complex?

13) Reviewer suggests that author to show the correlation between levels of tTG and Hsp90 in Bevacizumab sensitive and refractory PDX in regards in correlation to VEGF90K in figure 7.

14) Could author show existence of Hsp90 and tTG existence by adding a Western blot band to the panel in figure 7B? Is it possible to show any correlation between VEGF90K accumulation in the cells and levels of tTG and Hsp90 in the cells both in vitro and in vivo?

Minor comments:

Why the MDAMB231 cell lines were grafted subcutaneously and the PDX samples were implanted orthotopically?

Reviewer believes that the figure 2D-F can be removed due to the lack of critical information.

In the diagram presented as figure 8, there is no mention of tTG while it has the crucial role of cross-linking of the VEGF165. It would be very informative to include tTG in the diagram.

Reviewer #3 (Remarks to the Author):

Comments to the Author

In this work, the authors investigated a combination therapy using Bevacizumab and HSP90 inhibitor, and a mechanism of the effect of the combination therapy through cancer-derived microvesicles (MVs). This study needs to be improved by addressing the following points:

Specific comments

1. The authors discuss MVs and exosomes in result section. However, MVs and extracellular vesicles (EVs, include exosomes) are different about size, marker protein and the mechanisms of generation, like the authors showed in Fig. 3D. The authors should clearly define that you focused on MVs.
2. The authors should discuss about why had you focus on HSP90.
3. Through all figure in this article, lane No. had shown in the data of western blot. The authors should clearly indicate "lane No." and should change position lane No. in each figures.
4. In Figure 1, the authors showed the data of tumor growth after treatment of some reagents. The authors should show the data of tumor weight in Fig. 1A and 1C, and should show the data of picture of tumors in Fig. 1A and 1B. Furthermore, the authors should perform statistical analysis in Fig. 1C.
5. In Figure 2D, the authors showed the data of SEM of MDA-MB-231 cells. The authors suggested that the arrows show MVs. However, it cannot show whether MVs or not from this result. The authors should show a data of immunoelectron microscopy using anti-Flotillin2 antibody. And also, the author should confirm MVs but not exosomes by western blotting analysis using anti-Flotillin2, anti-CD63, anti-CD9 and anti- β -actin antibodies.
6. The authors showed an existence of VEGF or HSP90 and actin along with surface of MVs in Figure 3C, 5A and 6A. However, these data cannot show whether MVs or not. The authors should show the data of immunofluorescence stain using anti-flotillin2.
7. In Fig. 3B, the authors should show the data of western blot using anti-flotillin2 antibody.
8. In Fig. 3D, to confirm to exclude the contamination of exosomes, the authors should show the data of western blot using anti-CD9, anti- β -actin and anti-flotillin2 antibody.
9. In Fig. 4B, 4C, 4D, 4E and 4F, the authors should show the data of western blot using anti- β -actin and anti-GAPDH as loading control.
10. In Fig. 4C, the authors showed phosphorylation of VEGFR2 after treatment of MVs from VEGF knock downed MDA-MB-231 cells. Furthermore, the authors showed the data of western blot of MVs after treatment siRNA. It is anticipated that the experiment in Fig. 4C had been used MVs from MDA-MB-231 treated with siRNA#1, #2 or #3. The authors should clearly show that which cells you used.
11. In Fig. 5B, the authors showed existence of HSP90 in VEGF90k-contained MVs. The authors should perform same experiment with Fig.5B using MVs from SKBR3 cells, because the authors showed the existence of VEGF90k in MVs from SKBR3 cells but not HeLa cells in Fig. 3E. Furthermore, the author should show the data of western blot using anti-flotillin2, anti-HSP90 and anti-VEGF antibodies in "MV protein inputs".
12. In Fig. 7C, 7E and 7F, the authors should show the data of western blot using anti- β -actin and anti-GAPDH antibodies as loading control.
13. The authors showed the data of tumor growth after treatment of bevacizumab and/or 17-AAG in Fig. 7G. Tumor growth of HCI-003 had been suppressed by treatment. Have you checked HCL-006, -008, -009 and -011? On the other hand, Can't Tumor growth suppress when used HCI-005, -007, -010 and -013? The authors should perform statistical analysis for the data of Fig. 7G.
14. The authors should proofread, checking their English usage.

RE: Reviewers' comments:

Reviewer #1 (Remarks to the Author):

This is a well written and well executed study describing a novel mechanism of pro-angiogenic signalling which may, potentially, help to explain resistance to the anti-angiogenic drug bevacizumab in breast cancer. I think that the work is technically sound and the experimental data would appear to support main conclusions of the study.

As the authors point out, bevacizumab has failed to make an appreciable impact in the treatment of breast cancer. It is important that we understand why, because this may lead to new treatment options for targeting the tumour vasculature in breast cancer. Therefore, this manuscript addresses an important question.

Specifically, the authors show that breast cancer cells can secrete a 90 kDa species of VEGF that is formed due to the crosslinking activity of transglutaminase. Moreover, they show that this 90 kDa species of VEGF is coupled to extracellular vesicles (EV) by HSP90. Interestingly, whilst these EVs show potent pro-angiogenic activity both in vivo and in vitro, this pro-angiogenic activity is not blocked by bevacizumab. The authors propose that this may be one mechanism through which bevacizumab activity is limited in breast cancer. Moreover, they show that this resistance mechanism is overcome by combining bevacizumab with HSP90 inhibitors, because the HSP90 inhibitors trigger the release of 90 kDa VEGF from EVs - thus allowing bevacizumab free access to block VEGF. The authors therefore propose that VEGF and HSP90 inhibitors should be combined in patients.

I have the following comments:

(1) Page 7, line 141. When the authors say "the VEGF drug" I think they should replace this with the word "bevacizumab" or "anti-VEGF antibody" because the term "the VEGF drug" is imprecise. Same issue should be corrected on page 15 line 327.

--- We have done this in both instances (i.e. replacing "VEGF drug" with "anti-VEGF antibody"), as suggested by the reviewer.

(2) Page 10, line 219. The authors say "Bevacizumab and other anti-VEGF antibodies" when referring to Fig 4D. In this figure, they only test ONE other anti-VEGF antibody, but the sentence infers that more than one other anti-VEGF antibody was tested. I believe that this is misleading and needs to be corrected. Same issue should be corrected on page 15 line 338.

--- We have corrected this in the two lines suggested by the reviewer by stating "either Bevacizumab or a pan anti-VEGF antibody", instead of "Bevacizumab and other anti-VEGF antibodies".

(3) Page 15, line 326. The authors say that their mechanism "does not involve nor require the upregulation of VEGF in the targeted endothelial cells." However, I could not find the experimental evidence where the authors prove this? Could the authors make it clear what evidence they provide to support this statement? If there is no evidence presented in the paper, it may be better to remove this sentence from the manuscript.

--- The reviewer is correct that we cannot completely rule out this possibility. The initial stimulation of VEGF receptor-signaling by microvesicles with associated VEGF_{90k}, that occurs within a matter of minutes, can not be due to the upregulation of (endogenous) VEGF expression in endothelial cells. However, the more sustained stimulation could be contributed by endogenous VEGF in response to microvesicles activating the appropriate signaling pathways that up-regulate VEGF expression. Thus, we have removed the sentence in question.

(4) Page 16, line 348-349. I have a problem with the author's hypothesis that the trapping of VEGFR2 at the cell surface may lead to an increased pro-angiogenic signalling of VEGFR2. There are several studies showing that, in fact, the endocytosis of VEGFR2 is required for signal transduction to downstream pathways and the full activation of angiogenesis:

*Lanahan et al
VEGF receptor 2 endocytic trafficking regulates arterial morphogenesis.
PMID: 20434959*

*Sawamiphak et al
Ephrin-B2 regulates VEGFR2 function in developmental and tumour angiogenesis.
PMID: 20445540*

*Gourlaouen et al
Essential role for endocytosis in the growth factor-stimulated activation of ERK1/2 in endothelial cells.
PMID: 23341459*

Whilst it is fine for the authors to speculate that trapping at the cell surface protects the receptor from degradation / dephosphorylation, they should add a brief note to the Discussion acknowledging that endocytosis of VEGFR2 may in fact be required to deliver the full pro-angiogenic signal to endothelial cells (and these three papers above should be cited as evidence of this fact).

--- As requested by the reviewer, we now present this possibility in the "Discussion" of the revised manuscript (page 16, starting on line 363) and cite the papers suggested by the reviewer.

(5) Related to the above. Is it possible that the EVs are endocytosed more effectively by the endothelial cells (compared to free VEGF) or signal more effectively from endosomes in endothelial cells and that this explains the heightened signalling by EV-coupled 90 kDa VEGF? I do not propose that the authors perform further experiments to address this, but I do propose that some text regarding the potential for EVs to promote VEGFR2 signalling from endosomes should be added. However, the authors are free to ignore this comment if EVs are, in fact, too large to be effectively endocytosed.

--- While we suspect that MVs are too large to be endocytosed, we agree with the reviewer that MVs might trigger additional signaling pathways that lead to increased VEGF expression and autocrine signaling from intracellular sites (e.g. endosomes), as indicated in our answer to 3, above.

(6) Page 16, lines 360-361. The authors state correctly that there are other mechanisms of resistance to anti-VEGF therapy. Although reference 24 is a good one, it is now rather old and

there are more recent reviews of this rapidly moving field. In addition to citing reference 24, I suggest to also cite these additional more recent references that discuss mechanisms of resistance to anti-VEGF therapy (see below). I also suggest to remove reference 51, as this paper does not actually address the mechanisms of resistance to anti-VEGF drugs in breast cancer.

Ebos et al (2011)

Antiangiogenic therapy: impact on invasion, disease progression, and metastasis.

PMID:21364524

Vasudev et al (2014)

Anti-angiogenic therapy for cancer: current progress, unresolved questions and future directions.

PMID: 24482243

--- We now remove what was previously reference 51 and cite the references suggested by the reviewer in the revised manuscript (i.e. now references 54 and 55).

(7) Page 16 - 17. It is good to see that the authors discuss the limitations of their study and the questions that still remain unanswered, because these comments highlight important areas for further research. However, I propose that they need to include two further points within the Discussion section of the article:

Firstly, unless I am mistaken, they have not proven the presence of VEGF-90K vesicles in human tumours. I understand that they have shown their existence in cultured patient-derived cancer cells, but can these same vesicles be isolated from fresh human tumour specimens of breast cancer (without the culture step) or from the blood of humans with breast cancer? Authors should state that they did not demonstrate this in the current study and that this is something that should be examined in the future.

Secondly, the authors experiments only address the effect of the bevacizumab + HSP90 combination on subcutaneously implanted tumours (i.e. a model that roughly equates to primary breast cancer). However, several studies have now shown that, whilst anti-angiogenic therapies can be effective against models of primary breast cancer, the same anti-angiogenic therapies are very poorly effective (or sometimes completely ineffective) in models of metastatic breast cancer (e.g. refs. 56, 57 Ebos et al Cancer Cell PMID: 19249681, e.g. Guerin et al Cancer Research PMID: 23610448) Authors should state that they did not demonstrate efficacy of bevacizumab + HSP90 combination in metastatic breast cancer models in the current study and that this is something that should be examined in the future.

--- Yes, we agree with the reviewer about each of these points and now make it clear in the revised manuscript that it will be very important in the future to examine plasma and serum samples from cancer patients for the presence of microvesicles and associated VEGF_{90K}, as well as examine the combination of drug treatments in metastatic breast cancer models (see page 18, starting on line 406). Indeed, we are setting up both of these lines of study.

Reviewer #2 (Remarks to the Author):

The manuscript by Feng et al. deals with the interesting topic of Bevacizumab resistance in breast cancer. By using the triple negative breast cancer cell line MDAMB231, the authors show that VEGF₁₆₅ cross links with help of tTG enzyme and creates VEGF_{90K} which is subsequently packed into microvesicles with help of Hsp90 and is shed out of cells and cause resistance to Bevacizumab. The authors have validated their hypothesis using another breast cancer cell line SKBR3 and cervix adenocarcinoma HeLa cells as negative control. Results are further validated in responsive and refractory patient derived materials. Authors confirm the in vitro findings with in vivo tumor growth curves using tumor cell lines and also patient derived material however some more complementary experiments are required to shed light to this new concept of drug resistance.

1) Reviewer was wondering if the MVs containing VEGF_{90K} are only specific to breast cancer or if they can be also found in other cancer types? Could the author present some Western blots from other types of cancers that are also resistant to Bevacizumab in order to clarify the specificity of these vesicles to breast cancer.

--- MVs containing VEGF_{90K} are not specific to breast cancer. We have found MVs containing VEGF_{90K} in other cancer types that have shown resistance to Bevacizumab. Specifically, MVs isolated from the human glioblastoma U87 cell line, and from the human colorectal adenocarcinoma HT-29 cell line, also contain VEGF_{90K}. This is now stated on page 8, lines 179-181, and shown in Supplementary Fig. 3F, in the revised manuscript. In the future, we intend to examine whether various other cancer cells shed VEGF_{90K}.

2) Reviewer would like the author to be more thorough about the nature of the cross-linked VEGF_{90K}. Why and how does the cross-linking of VEGF create the 90K VEGF and not any other molecular weight. Do the authors suggest that there is always a very defined number of crosslinked VEGF molecules? Furthermore, can the authors rule out that the found protein of 90 kDa is a result of crosslinking VEGF-A and Hsp90 or other VEGF-A interacting proteins?

--- We have found that the cross-linking of VEGF also yields crosslinked products recognized by anti-VEGF antibodies with apparent molecular mass of 70 kDa and 110 kDa. However, thus far, we have only detected the 90 kDa VEGF immunoreactive band (VEGF_{90K}) in microvesicles.

When we first found that both pan-VEGF and anti-VEGF₁₆₅ antibodies recognize a specific 90 kDa protein band, we performed immunoprecipitation (IP) assays with these antibodies. The immunoprecipitated 90 kDa protein band was excised and subjected to microsequencing at the Cornell Proteomics Facility. This analysis indicated that the 90 kDa immunoprecipitated protein band contained VEGF-A and Hsp90. Additional IP experiments further showed that VEGF and Hsp90 form a complex; however, we do not believe that VEGF_{90K} is the result of crosslinking VEGF-A to Hsp90, as we have found that VEGF_{90K} can be generated *in vitro* from recombinant VEGF₁₆₅ (Figs. 5C, 6G and 6H), whereas, we have not been able to detect VEGF₁₆₅ covalently crosslinked to Hsp90.

3) Due to the correlation between the intra-tumoral hypoxia and VEGF levels in tumor microenvironment, reviewer suggests to show hypoxic regions under different conditions to be displayed in supplementary figure S1 A. Furthermore, the authors should analyze basic features

of treated and control tumors, such as hypoxic regions, blood vessel density, blood vessel perfusion, and metastatic index.

--- In addressing these points, I should begin by saying that from our perspective, the initial experiments with MDAMB231 xenografts and breast cancer PDXs served as a launching point for identifying what appeared to be an interesting relationship between Bevacizumab and the Hsp90 inhibitor 17AAG. This led us to discover an unexpected mechanism by which microvesicles containing a unique form of VEGF, bound to Hsp90, gave rise to a potent activation of VEGF receptor-signaling in a manner that was insensitive to Bevacizumab. The discovery and characterization of these microvesicles represents the thrust of this initial study. There is now a good deal we would like to do in subsequent studies to further characterize the potential therapeutic benefits of combining Bevacizumab with Hsp90 inhibitors like 17AAG. What we have done thus far is to count the blood vessels and endothelial cells for the different treatments being examined (Supplementary Figs. 1A and 1B), and we have used an HIF-1 α antibody in immunohistochemical staining to show hypoxic regions under different conditions. These latter experiments showed that the combination of Bevacizumab and 17AAG treatment of mice implanted with HCI-001 tumors induced hypoxia, now presented in Supplementary Fig. 1A (bottom panels) of the revised manuscript. We further examined whether microvesicle-associated VEGF_{90K} secretion was increased under hypoxic conditions, using CoCl₂ to mimic hypoxia in cell culture. When using a VEGF ELISA assay, we found that these conditions enhance total VEGF secretion from both MDAMB231 cells and HCI-002 cells (shown in the right-hand plots in Reviewer Figs. 1A and 1B, included here for the reviewer). However, such conditions did not appear to influence the biogenesis of microvesicles nor the amount of microvesicle-associated VEGF_{90K} secreted from these tumor cells (left-hand panels in Reviewer Figs. 1A and 1B, included here for the reviewer).

We also have collected data for the survival of MDAMB231 and HCI-002 xenografts under the different treatment conditions. In these experiments, all of the mice treated with the combination of Bevacizumab and 17AAG survived through 28 days, whereas 80-100% of the control mice had to be sacrificed within that time period because of the growth of large tumors in the animals, and the same was true for the mice treated with just one or the other of the drugs. However, we recognize that mouse models for breast cancer metastasis have often yielded disappointing results when attempting to block tumor angiogenesis (also see Reviewer 1, point 7, above). Consequently, we are currently in the process of performing experiments specifically examining lung metastasis, using this combination treatment approach. Developing these models represents a significant undertaking and we are planning to make this a separate study.

4) Due to the vast physiological functions of Hsp90, how can the author exclude the function of 17AAG treatment merely on the Hsp90s binding to the VEGF90K and not all the endogenous Hsp90s present within the cell? How can the authors show that the inhibition with 17AAG, does not affect and interfere with the general physiological functions of the Hsp90 protein?

--- The reviewer is correct that we cannot exclude the possible impact of 17AAG treatment on the general physiological functions of the Hsp90 protein (e.g. its potential effects on HIF/VEGF signaling, endothelial cell survival and tumor angiogenesis). We now mention this in the "Discussion" of the revised manuscript (page 17, lines 376-379). Still, we found that 17AAG treatment alone did not significantly inhibit tumor growth in mice (Figs. 1A-C in the revised manuscript). Moreover, it exerted only a limited influence on the

ability of MDAMB231 cells to recruit endothelial cells into angioreactors implanted into mice (Reviewer Fig. 2, included here for the reviewer).

5) *Regarding figure 2G, reviewer suggests adding another control group containing only rVEGF165+17AAG in order to investigate if there would still be a formation and accumulation of VEGF90K in the presence of tTG without Hsp90? And as a follow up experiment, in figure 2I, the reviewer suggests to add a control group containing HUVEC cells cultured together with rVEGF165 +17AAG.*

--- In the angioreactor experiments, there is no tTG present when adding rVEGF₁₆₅, and so there is no production of VEGF_{90K}. As indicated above, we have shown that 17AAG alone can not effectively block the MDAMB231 cell-stimulated migration of endothelial cells into angioreactors implanted into mice. As requested by the reviewer, we have also added the data for HUVECs treated with rVEGF₁₆₅ and 17AAG in the tubulogenesis assays (Supplementary Fig. 2C in the revised manuscript).

6) *Reviewer believes that the image in figure 3C could be improved by using a more specific cell surface marker or using a cell membrane marker (dye) to visualize the cell surface together with Actin specific marker with better visual quality to show the MVs containing VEGF90K.*

--- As suggested by the reviewers, we have used the lipid-binding dye FM 1-43FX to stain MVs on the cell surface (Supplementary Fig. 3C in the revised manuscript). We also used an anti-flotilin-2 antibody as a membrane marker to visualize MVs on MDAMB231 cells (Supplementary Fig. 3D in the revised manuscript).

7) *In order to clarifying the specific role of tTG in cross linking and generating the VEGF90KD, it is necessary to include a Western Blot band in figure 3H to show the lack of VEGF90K upon knock down of the tTG or in presence (control) of the enzyme.*

--- This Western Blot has been added to Figure 3H (third lane from the top).

8) *In order to show the specific function of tTG in cross-linking the VEGF165 and create VEGF90K prior to it's binding to Hsp90, reviewer recommends to inhibit Hsp90 and show that no VEGF90K will be found in the shedding MVs. Also, to show the predicted accumulation of VEGF90K in the cytoplasm upon inhibition of Hsp90, it is suggested to perform a Western Blot on the cell lysate from the cells treated with 17AAG.*

--- We show that MDAMB231 cells treated with the Hsp90 inhibitor 17AAG yielded MVs that no longer contained VEGF (this is shown in Fig. 6A, and perhaps best in the far-right insert panels in the revised manuscript).

We also examined the levels of VEGF_{90K} in cell lysates following treatment of the cells with 17AAG. After 30 minutes of 17AAG treatment, we observed an ~2 fold increase in the level of VEGF_{90K} when analyzing cell lysates (Reviewer Fig. 3, included here for the reviewer). However, this accumulation was not sustained, with the VEGF_{90K} levels returning to those of the control after 90 minutes. We do not yet completely understand why the return to control levels occurs. One speculative possibility is that initially treatment with the Hsp90 inhibitor prevents VEGF_{90K} from associating with Hsp90 along the surfaces of MVs, which represents the predominant mechanism by which VEGF_{90K} is secreted from cells, thereby leading to its accumulation in the whole cell lysate preparations. However, with extended time periods, free VEGF_{90K} (i.e. not bound to

Hsp90) may be secreted through an alternative (e.g. classical) mechanism thus preventing its accumulation in cell lysates.

9) *Due to increase in VEGF165 levels under hypoxic conditions, do the authors expect increased levels of VEGF90K also in these conditions?*

In figure 4B, when on the right panel, MDAMB231 MVs, why the levels of VEGF90K does not go down after 45 min? Has the author checked longer time points? Is there any accumulation of the VEGF90K which leads to stabilizing the protein after 45 min?

--- We would have expected increased levels of VEGF_{90K} under hypoxic conditions. However, our data shows that hypoxia does not influence MV-associated VEGF_{90K} secretion from these tumor cells (Reviewer Figs. 1A and 1B, left panels, included here for the reviewer), but apparently hypoxia significantly increases the secretion of the smaller classical forms of VEGF (Reviewer Figs. 1A and 1B, right panels). We suspect that the levels of activated tTG or MV biogenesis may be the bottleneck, i.e. in terms of why the secretion of MV-associated VEGF_{90K} is not increased.

With regard to the question raised concerning Figure 4B, we believe that what the reviewer is referring to are the levels of VEGF receptor activation and signaling activity (i.e. in Figure 4B). We have checked longer time points and see that the VEGF receptor-signaling is sustained through 90 minutes. This is described on page 10, lines 221-223, and shown in Supplementary Fig. 4B in the revised manuscript.

10) *Reviewer suggests that the authors add two negative controls for figure 5A and 6A, i) VEGF KD; ii) Incubate the sample with IgG control and secondary antibody to eliminate the possibility of an unspecific secondary antibody signal.*

--- The requested negative controls have now been included in the revised manuscript. The control for the VEGF knock-down is described on page 11, lines 245-246, and shown in Supplementary Fig. 5A of the revised manuscript. In this control experiment, conditions were used to achieve a partial knock-down of VEGF so as to directly compare the anti-VEGF antibody staining of cells still expressing VEGF ('a' arrows in Supplementary Fig. 5A) versus cells that are not expressing VEGF ('b' arrows). Supplementary Fig. 6 shows the secondary antibody controls (stated on page 12, line 272 of the revised manuscript).

11) *Regarding figure 5C, in order to show the crosslinking effect of tTG on VEGF165, reviewer wish to suggest an additional control as following:*

rVEGF165 + rtTG + rHsp90 - (basically rVEGF165 + rtTG in absence of rHsp90).

--- The additional control is described on page 12, lines 256-259, and shown in Supplementary Fig. 5B of the revised manuscript.

12) *Reviewer wonders why in figure 6G, there is no stimulation of VEGFR2 when HUVEC cell lysate is incubated with VEGF90K-Hsp90 complex?*

--- We have found that free VEGF_{90K}-Hsp90 complex (without MVs) can stimulate VEGFR2 autophosphorylation (Fig. 6G, top panel, lane 1); however, it is fully sensitive to Bevacizumab (lane 2). It is only when VEGF_{90K} is associated with microvesicles via Hsp90 that it is insensitive to Bevacizumab.

13) Reviewer suggests that author to show the correlation between levels of tTG and Hsp90 in Bevacizumab sensitive and refractory PDX in regards in correlation to VEGF90K in figure 7.

--- (see below)

14) Could author show existence of Hsp90 and tTG existence by adding a Western blot band to the panel in figure 7B? Is it possible to show any correlation between VEGF90K accumulation in the cells and levels of tTG and Hsp90 in the cells both in vitro and in vivo?

--- To address points 13 and 14 raised by this reviewer, we now show the levels of tTG and Hsp90 in the whole cell lysates from tumor samples HCI-001 to HCI-003, and HCI-005 to HCI-010 in Supplementary Fig. 7 of the revised manuscript. Thus far, we are hesitant to try to draw any strict correlations between the levels of tTG and Hsp90 in cancer cells, and either the amounts of VEGF_{90K} in cell lysates or secreted in microvesicles, particularly because we have not yet quantified the levels of activated tTG in these cases. What we can say is that those cells that secrete VEGF_{90K} contain these two other key proteins. We also now mention the potential importance of the levels of tTG and Hsp90 in cancer cells with regard to their abilities to generate and shed microvesicles with associated VEGF_{90K} in the 'Discussion' of the revised manuscript (page 18, lines 399-401).

Similarly, the amounts of VEGF_{90K} in cell lysates do not always strictly correlate with the levels of secreted VEGF_{90K}, because the latter is dependent upon the ability of the different PDXs to generate microvesicles. We make reference to this in the 'Discussion' of the revised manuscript (page 18, lines 394-401).

Minor comments:

Why the MDAMB231 cell lines were grafted subcutaneously and the PDX samples were implanted orthotopically?

--- In our initial sets of experiments, MDAMB231 cells were grafted subcutaneously, i.e. upon introducing the cells with Matrix gel. However, when we began experiments with the PDX samples, we simply felt the more appropriate method would be to implant the cells orthotopically by surgery.

Reviewer believes that the figure 2D-F can be removed due to the lack of critical information.

--- We have considered the reviewer's request; however, we would still prefer to include these figures in the manuscript, as they help to further emphasize that we are working with microvesicles rather than exosomes (i.e. a point raised by Reviewer 3, below).

In the diagram presented as figure 8, there is no mention of tTG while it has the crucial role of cross-linking of the VEGF165. It would be very informative to include tTG in the diagram.

--- We agree that ideally it would be informative to include tTG in the diagram. However, we have given this a good deal of thought and have decided to leave tTG out of the figure, for purposes of clarity, but to indicate its role in the figure legend.

Reviewer #3 (Remarks to the Author):

Comments to the Author

In this work, the authors investigated a combination therapy using Bevacizumab and HSP90 inhibitor, and a mechanism of the effect of the combination therapy through cancer-derived microvesicles (MVs). This study needs to be improved by addressing the following points:

Specific comments

1. The authors discuss MVs and exosomes in result section. However, MVs and extracellular vesicles (EVs, include exosomes) are different about size, marker protein and the mechanisms of generation, like the authors showed in Fig. 3D. The authors should clearly define that you focused on MVs.

--- We have tried to emphasize throughout the revised manuscript that we are focusing on actin-based larger MVs (see page 3, starting with line 56; page 5, starting with line 112). We also explain in some detail how we are isolating MVs from cancer cells (page 6, starting with line 117).

2. The authors should discuss about why had you focus on HSP90.

--- As stated on page 4 of the revised manuscript, upon examining different drug combinations in mouse models for breast cancer, we found that combining Bevacizumab with an Hsp90 inhibitor gave particularly effective results. Thus, we were interested in how Hsp90 might fit into our findings regarding MV-stimulated tumor angiogenesis and Bevacizumab-resistance. Moreover, we knew that Hsp90 is a MV-associated protein. We then showed that Hsp90 is a VEGF-interacting protein by immunoprecipitation (IP) assays using VEGF antibodies followed by mass spectrometry. We further demonstrated that Hsp90 binds VEGF_{90K}. It is for these reasons that we focused on Hsp90. Also, please see the response to reviewer 2, point 2.

3. Through all figure in this article, lane No. had shown in the data of western blot. The authors should clearly indicate "lane No." and should change position lane No. in each figures.

--- We have now indicated the lane #s for each Western blot in the revised manuscript.

4. In Figure 1, the authors showed the data of tumor growth after treatment of some reagents. The authors should show the data of tumor weight in Fig. 1A and 1C, and should show the data of picture of tumors in Fig. 1A and 1B. Furthermore, the authors should perform statistical analysis in Fig. 1C.

--- We now present the data for the tumor weights for Figs. 1A and 1C, as well as the statistical analysis for each of the figures (Figs. 1A-C in the revised manuscript). We also show as examples, the pictures of the tumors for the patient-derived tumor grafts in Figs. 1B and 1C. We do not show the tumor growth for the MDAMB231 xenografts in Fig. 1A because the tumors for the control and individual treatments looked identical, whereas, we could not detect any tumors with the combination treatment.

5. In Figure 2D, the authors showed the data of SEM of MDA-MB-231 cells. The authors suggested that the arrows show MVs. However, it cannot show whether MVs or not from this result. The authors should show a data of immunoelectron microscopy using anti-Flotillin2

antibody. And also, the author should confirm MVs but not exosomes by western blotting analysis using anti-Flotillin2, anti-CD63, anti-CD9 and anti-βactin antibodies.

--- We show the picture in Figure 2D because MDAMB231 cells are highly effective at generating these large EVs, visualized by EM, which match the size of the MVs shed from cancer cells, as determined by immunofluorescence when staining for actin or using the lipid-binding dye FMIX-43FX. We have used similar images for this purpose in other publications (e.g. Antonyak et al., PNAS, 2011, 108, 17569; Desrochers et al., Nat Commun, 2016, 11, 11958). We have also performed immunofluorescence with anti-flotillin-2 antibody to visualize microvesicles (MVs) on the surfaces of cancer cells. This is now stated on page 8, lines 167-170, and presented in Supplementary Fig. 3D of the revised manuscript.

We have confirmed the isolation of MVs (i.e. resolved from exosomes) by Western blotting analysis, using anti-CD63 to detect exosomes, and anti-actin antibody to detect MVs, as shown in Fig. 3D and described on page 8, starting on line 170 of the revised manuscript. Flotillin-2 is a marker for both of these classes of extracellular vesicles.

6. The authors showed an existence of VEGF or HSP90 and actin along with surface of MVs in Figure 3C, 5A and 6A. However, these data cannot show whether MVs or not. The authors should show the data of immunofluorescence stain using anti-flotillin2.

--- We have also used anti-flotillin-2 antibody as a membrane marker to visualize MVs on MDAMB231 cells (Supplementary Fig. 3D of the revised manuscript).

7. In Fig. 3B, the authors should show the data of western blot using anti-flotillin2 antibody.

--- Fig. 3B represents total secreted proteins. We show anti-flotillin staining when examining the MVs shed from cancer cells (e.g. Figs. 3E, 3F, and 3H, of the revised manuscript).

8. In Fig. 3D, to confirm to exclude the contamination of exosomes, the authors should show the data of western blot using anti-CD9, anti-βactin and anti-flotillin2 antibody.

--- We have confirmed that we are resolving MVs from exosomes by Western blotting analysis using anti-CD63 (exosomes) and anti-actin antibodies (MVs) (Fig. 3D in the revised manuscript). Flotillin-2 is present in both classes of extracellular vesicles.

9. In Fig. 4B, 4C, 4D, 4E and 4F, the authors should show the data of western blot using anti-βactin and anti-GAPDH as loading control.

--- The data for Western blots using anti-actin antibody has now been added to each of these figures in the revised manuscript.

10. In Fig. 4C, the authors showed phosphorylation of VEGFR2 after treatment of MVs from VEGF knock down MDA-MB-231 cells. Furthermore, the authors showed the data of western blot of MVs after treatment siRNA. It is anticipated that the experiment in Fig. 4C had been used MVs from MDA-MB-231 treated with siRNA#1, #2 or #3. The authors should clearly show that which cells you used.

--- Yes, HUVEC cells were treated with MVs from MDAMB231 cells first treated with siRNAs targeting VEGF. We state this on page 10, lines 224-225 of the revised manuscript.

11. In Fig. 5B, the authors showed existence of HSP90 in VEGF90k-contained MVs. The authors should perform same experiment with Fig.5B using MVs from SKBR3 cells, because the authors showed the existence of VEGF90k in MVs from SKBR3 cells but not HeLa cells in Fig. 3E. Furthermore, the author should show the data of western blot using anti-flotillin2, anti-HSP90 and anti-VEGF antibodies in "MV protein inputs".

--- The same experiment as presented in Fig. 5B, but using MVs from SKBR3 cells, has been performed (now shown in Supplementary Fig. 5B of the revised manuscript). The data from Western blot analyses using anti-flotillin-2, anti-HSP90 and anti-VEGF antibodies in "MV protein inputs" is now shown in Fig. 5B of the revised manuscript.

12. In Fig. 7C, 7E and 7F, the authors should show the data of western blot using anti- β actin and anti-GAPDH antibodies as loading control.

--- The Western blots using an anti-actin antibody have been added to the indicated figures in the revised manuscript.

13. The authors showed the data of tumor growth after treatment of bevacizumab and/or 17-AAG in Fig. 7G. Tumor growth of HCI-003 had been suppressed by treatment. Have you checked HCL-006, -008, -009 and -011? On the other hand, Can't Tumor growth suppress when used HCI-005, -007, -010 and -013? The authors should perform statistical analysis for the data of Fig. 7G.

--- We have thus far checked HCI-005 and HCI-009. HCI-005 is Bevacizumab-insensitive, whereas, HCI-009 is suppressed by Bevacizumab treatment. Statistical analyses for the data in Fig. 7G have now been added.

14. The authors should proofread, checking their English usage.

--- We and a number of colleagues have proofread the manuscript.

Reviewer Fig 1

A.

B.

Fig 1. A. MDAMB231 cells were cultured in CoCl₂ (100 μM) containing medium (hypoxia) or medium without CoCl₂ (normoxia). Microvesicles (MVs) prepared from the same amount of hypoxia or normoxia MDAMB231 cells were immunoblotted with antibodies against VEGF and the extracellular vesicle (EV) marker flotillin-2 (Left, top two panels). Lysates of the hypoxia or normoxia MDAMB231 cells were immunoblotted with antibodies against HIF-1α and actin (Left, bottom two panels). Right plot: Relative VEGF secretion from the hypoxia or normoxia MDAMB231 cells were measured by VEGF ELISA. **B.** HCI-002 primary tumor cells were cultured in CoCl₂ (100 μM) containing medium (hypoxia) or medium without CoCl₂ (normoxia). MVs prepared from the same amount of hypoxia or normoxia HCI-002 primary tumor cells were immunoblotted with antibodies against VEGF and flotillin-2 (Left, top two panels). Lysates of the hypoxia or normoxia HCI-002 primary tumor cells were immunoblotted with antibodies against HIF-1α and actin (Left, bottom two panels). Right plot: Relative VEGF secretion from the hypoxia or normoxia HCI-002 primary tumor cells was measured by VEGF ELISA.

Reviewer Fig 2

Relative stimulation of angiogenesis in nude mice

Samples	Control	MDAMB231 cells	MDAMB231 cells + anti-VEGF	MDAMB231 cells + 17AAG
Relative fold	1.000 ± 0.201	10.97 ± 1.785	8.155 ± 2.580	9.72 ± 2.680

Fig 2. The relative amounts of endothelial cells that entered implanted angioreactors that lacked any activators (vehicle control, histogram 1), or were loaded with MDAMB231 cells (5×10^4 cells/angioreactor) without (histogram 2) or with a pan inactivating VEGF antibody (200 ng/angioreactor; histogram 3), or were loaded with 10 μ M 17AAG (histogram 4). Further quantification of the experimental results is shown in the bottom table.

Reviewer Fig 3

Fig 3. MDAMB231 cells were lysed after being exposed to 17AAG for the indicated periods of time (lanes 1-4). The lysates were then immunoblotted with antibodies that recognize VEGF or actin.

Reviewers' comments:

Reviewer #1 (Remarks to the Author):

The authors have addressed all of my comments to my satisfaction.

Reviewer #2 (Remarks to the Author):

We believe that the manuscript "A class of extracellular vesicles from breast cancer cells uniquely activates VEGF receptors and tumor angiogenesis" by Feng et al. has greatly improved and that by now most of our questions were answered sufficiently. However we still have some remarks about the current manuscript:

1. We still have major concerns about the diagram summarizing the findings of this paper (Fig. 8):

- First of all, since it is one of the central parts of this story, please include tTG into the summarizing cartoon. The argument of having it left out "for purposes of clarity" does not appeal to us.
- Why do the authors put VEGF90kDa on the surface of the MVs? To our understanding it should be inside of the vesicles and only thereby being protected from VEGF-inhibitors like Bevacizumab. This actually makes sense if these vesicles fuse with VEGFR-2 containing vesicles within endothelial cells and thereby activate this pathway intracellular (possibly in a similar way as endogenous VEGF-A in endothelial cells) protected from extracellular VEGF-inhibitors.

2. Why did the authors show many of the requested data only as figures for the reviewer and did not include them into the manuscript? Since they increase the clarity and quality of this paper, we would strongly recommend including them into the shown supplementary figures.

3. The authors claim that they also found other forms of crosslinked VEGF-A (70kDa and 110kDa), however not in MVs. Please include the according Western Blots supporting this statement.

4. Although we agree that it is beyond the scope of this paper to fully address the complete in vivo characterization using multiple combinations of inhibitors as well as different tumor models, at least the requested metastasis quantification in the provided settings of tumor growth (Fig. 1) should have been provided. Especially in the light of strongly enhancing hypoxia in their tumor samples, a very well known condition linked to enhanced cancer cell spreading.

Reviewer #3 (Remarks to the Author):

In the revised manuscript submitted by Feng et al., the authors have provided satisfactory responses to my concerns. Therefore, I feel that the current version is suitable for publication in Nature Communications.

Reviewer #2 (Remarks to the Author):

We believe that the manuscript “A class of extracellular vesicles from breast cancer cells uniquely activates VEGF receptors and tumor angiogenesis” by Feng et al. has greatly improved and that by now most of our questions were answered sufficiently. However we still have some remarks about the current manuscript:

1. We still have major concerns about the diagram summarizing the findings of this paper (Fig. 8):

- First of all, since it is one of the central parts of this story, please include tTG into the summarizing cartoon. The argument of having it left out “for purposes of clarity” does not appeal to us.

- Why do the authors put VEGF90kDa on the surface of the MVs? To our understanding it should be inside of the vesicles and only thereby being protected from VEGF-inhibitors like Bevacizumab. This actually makes sense if these vesicles fuse with VEGFR-2 containing vesicles within endothelial cells and thereby activate this pathway intracellular (possibly in a similar way as endogenous VEGF-A in endothelial cells) protected from extracellular VEGF-inhibitors.

As requested by this reviewer, we now include tTG in one of our summary schematic diagrams (Figure 8A in the revised manuscript). We believe that tTG most likely catalyzes the crosslinking of VEGF (e.g. VEGF₁₆₅) when both of these proteins are accessible to the outside of the cell, where tTG is most active (we state this in the ‘Discussion’, on page 16, lines 349-351, of the revised manuscript). We suspect that VEGF₁₆₅ is delivered to the outer surfaces of cells through a classical secretory pathway, where it encounters tTG on the maturing microvesicles. This would be consistent with our finding that brefeldin A blocks the formation of VEGF_{90K} (stated in the ‘Discussion’ of the revised manuscript on page 19, lines 416-421). It also represents some of the reasoning that leads us to put VEGF_{90K} on the surface of MVs. Moreover, this idea is further supported by our findings that MVs can rapidly activate VEGF receptors when they are added to endothelial cells and from experiments in which we inhibited MV-associated Hsp90. In the latter case, we found that when MVs isolated from MDAMB231 cells were treated with 17AAG, and then collected again on 0.22 micron filters, the flow-through from the filtration contained the majority of the VEGF_{90K} due to its dissociation from the MVs (Figure 6D, compare the top and bottom panels; also, discussed on page 13, lines 282-289 of the revised manuscript). This experiment would seem to indicate that VEGF_{90K} is on the outside surface of the MVs, where it is associated with Hsp90 and has a weakened affinity for VEGF antibodies like Bevacizumab, compared to VEGF_{90K} not associated with MVs. Treatment of the MVs with 17AAG results in the rapid dissociation of VEGF_{90K} from the vesicles, at which point it can bind Bevacizumab with high affinity.

2. Why did the authors show many of the requested data only as figures for the reviewer and did not include them into the manuscript? Since they increase the clarity and quality of this paper, we would strongly recommend including them into the shown supplementary figures.

As requested by the reviewer, we now have added to the manuscript each of the figures that we had originally included for the reviewers only. Reviewer Figures 1A and 1B are now Supplementary Figures 8A and 8B. These are discussed in the 'Discussion' of the revised manuscript (page 16, lines 351-356). Reviewer Figure 2 is now Supplementary Figure 2A and is referred to in the 'Results' of the revised manuscript (page 5, lines 104-109). Reviewer Figure 3 is now Supplementary Figure 8C and is discussed in the 'Discussion' on page 16 (lines 356-363) of the revised manuscript.

3. The authors claim that they also found other forms of crosslinked VEGF-A (70kDa and 110kDa), however not in MVs. Please include the according Western Blots supporting this statement.

We now show the Western Blots supporting this statement in Supplementary Figure 3C. The additional higher molecular mass forms of crosslinked VEGF-A (~70 kDa and 110 kDa) are marked by an asterisk star in lane 1 of this figure (also discussed on page 7, lines 157-160 in the revised manuscript).

4. Although we agree that it is beyond the scope of this paper to fully address the complete *in vivo* characterization using multiple combinations of inhibitors as well as different tumor models, at least the requested metastasis quantification in the provided settings of tumor growth (Fig. 1) should have been provided. Especially in the light of strongly enhancing hypoxia in their tumor samples, a very well known condition linked to enhanced cancer cell spreading.

Based on what the reviewer raised in the previous critique, we have attempted to examine metastasis in the PDX HCI-001 mice used in the current study, specifically, by examining H&E staining with help from the Cornell Pathology Facility. Thus far, we have only been able to detect tumor cells in the lungs of the control animals; however, they are at the very early stage of forming metastatic nodules. A significant challenge has been that the PDX tumor grafts used in the experiments grow fast and thus the animals often need to be sacrificed before we have been able to reliably detect metastasis. Therefore, we are not able to draw any meaningful conclusion regarding whether Bevacizumab plus 17AAG treatment significantly affects metastasis in these animals. We also worry that if we are able to identify metastatic nodules in our PDX models, any interpretation might be ambiguous because of the effects of the combined treatment on primary tumor growth. For example, we might see less metastasis with the combination drug treatment because of a reduction in the size of the

primary tumors. In order to have any chance of properly assessing the effects of combining 17AAG and Bevacizumab on metastasis, using the PDX mice described in our current study, a number of additional experiments would be required where we harvest tumors from the different treatment groups at several time points to assess metastasis as a function of tumor size. However, again this would be dependent upon our being able to establish a proper time window in which we are able to detect metastatic colonies before sacrificing the animals. We also recognize that there are other mouse models that have been used for examining metastasis, and that experiments performed with these have raised questions regarding the effectiveness of anti-angiogenesis therapy (stated in the 'Discussion' of the revised manuscript, page 19, lines 423-425). Thus, we are now more inclined toward ultimately establishing these other models in the laboratory to assess the benefits or deficiencies of the combination of Bevacizumab with an Hsp90 inhibitor.

REVIEWERS' COMMENTS:

Reviewer #2 (Remarks to the Author):

The authors have clearly addressed all my remaining concerns. I think the manuscript is now suitable for publication in Nature Communications.